# Classification of tropical cyclone containing images using a convolutional neural network: performance and sensitivity to the learning dataset

Sébastien Gardoll[1] and Olivier Boucher[1]

[1]Institut Pierre-Simon Laplace, Sorbonne Université / CNRS, Paris, France

**Correspondence:** Sébastien Gardoll (sebastien.gardoll@cnrs.fr)

**Abstract.**

Tropical cyclones (TCs) are one of the most devastating natural disasters, which justifies monitoring and prediction on short and long timescales in the context of a changing climate. In this study, we have adapted and tested a convolutional neural network (CNN) for the classification of reanalysis outputs according to the presence or absence of TCs. This study compares the performance and sensitivity of a CNN to the learning dataset. For this purpose, we chose two meteorological reanalysis, ERA5 and MERRA-2, and used a number of meteorological variables from them to form TC-containing and background images. The presence of TCs is labelled from the HURDAT2 dataset. Special attention was paid on the design of the background image set to make sure it samples similar locations and times to the TC-containing images. We have assessed the performance of the CNN using accuracy, but also the more objective AUC and AUPRC metrics. Many failed classifications can be explained by the meteorological context, such as a situation with cyclonic activity, but not yet classified as TC by HURDAT2. We also tested the impact of spatial interpolation and of "mix and match" the training and test image sets on the performance of the CNN. We showed that applying an ERA5 trained CNN on MERRA-2 images works better than applying a MERRA-2 trained CNN on ERA5 images.

## 1 Introduction

Tropical cyclones (TCs) are localized, very intense circular low-pressure systems that form over warm tropical oceans and are associated with strong winds and heavy rains. They represent a major hazard for life and property in exposed regions of the world. There are still many unanswered questions on the number, intensity, duration, trajectory and probability of landfall of tropical cyclones in a warming climate (Emanuel, 2005; Webster et al., 2005; Chan, 2006; Vecchi et al., 2019; IPCC, 2021; Wu et al., 2022). IPCC (2021) estimated that "it is likely that the global proportion of major (Category 3–5) tropical cyclone occurrence has increased over the last four decades", but "there is low confidence in long-term (multi-decadal to centennial) trends in the frequency of all-category tropical cyclones". It has also been shown that global warming causes TCs to move further north in the North Atlantic and North Pacific basins (Kossin et al., 2014; IPCC, 2021; Studholme et al., 2021), which could have dire consequences for some coastal cities.

Better modelling of TCs in climate models is a prerequisite to estimate future changes in risk and associated damages. The automatic detection of TCs in climate model outputs is central to our ability to analyze results from climate projections. Indeed, TCs can only be simulated in models with sufficient horizontal and vertical resolutions (Strachan et al., 2013; Knutson et al., 2015; Vecchi et al., 2019; Roberts et al., 2020; Jiaxiang et al., 2020; Bourdin et al., 2022). Such models are now commonplace, but they produce huge volumes of output data. Furthermore, multiple long simulations are required because we need to understand the respective roles of decadal variability and climate trends in observed and simulated changes. It is, thus, important to have the capability to analyze climate simulations in a very efficient manner.

Climate modellers have developed "physical algorithms" to detect TCs based on the translation of their physical characteristics into identification criteria (e.g., Walsh et al., 2007; Horn et al., 2014; Bosler et al., 2016; Singh et al., 2022). Such detection algorithms generally rely on the identification of a spatial feature typical of a TC at all available time steps and a temporal correlation procedure to track the time consistency of the detected features and establish a trajectory. They are usually applied in predefined regions prone to TCs though it is not unusual for a TC to move outside its natural domain, hence it is important to apply the algorithms on a larger domain. These physical algorithms require to set up a number of thresholds which may depend on the climate model being considered and its resolution. For example, in the Stride Search algorithm (Bosler et al., 2016), a TC is identified if four criteria are met: maximum vorticity above a threshold, distance between the gridpoints of maximum vorticity and minimum sea level pressure below a threshold, the presence of a maximum vertically averaged temperature larger than its environment, and distance between the gridpoints of maximum vertically averaged temperature and minimum sea level pressure below a threshold. Bourdin et al. (2022) performed an intercomparison of four cyclone trajectory detectors –called trackers– on ERA5 reanalysis.

There is also a wealth of studies on the detection of TCs in satellite imagery, reanalysis and climate model outputs based on machine learning (ML) approaches. Table 1 summarizes notable studies published in the last eight years that implement neural architectures based on convolution layers. It is not surprising that this approach was favored because TCs have very distinct features which make them relatively easy to detect with convolutional neural networks. Since Liu et al. (2016), whose deep learning (DL) model only classifies patches of cyclone (i.e., small images centered on a cyclone), various subsequent studies have focused on improving the detection of all cyclones at once present in unidimensional or multidimensional meteorological images (e.g., Ebert-Uphoff and Hilburn, 2020) and climate model data (e.g., Matsuoka et al., 2018). This latter work focuses on the detection of cyclones using a CNN image classifier which operates on a sliding window of output from the Nonhydrostatic Icosahedral Atmospheric Model (NICAM) and studies the system performance in terms of detectability. The detection can be either "coarse" by drawing rectangular envelopes around the cyclones (the studies are flagged as detection in the Purpose column of Table 1) or "precise" by drawing the contours of the cyclones including their internal structure (studies flagged as segmentation). The main idea of these more recent studies is to apply new DL model architectures coming from computer vision research (e.g., U-Net, DeepLabv3, YOLOv3, Single Shot Detector, etc.) to the analysis of meteorological features such as cyclones. Most approaches for TC detection use supervised methods which require a training dataset. While such techniques are now mainstream, they are not always well documented and their description may lack sufficient details which are often key in ML, e.g., the data engineering involved in the preparation of the training dataset, the hyperparameters of the CNN and

the evaluation methods of the metrics used to measure the performance of the models. Studies evaluating the performance and sensitivity of TC detection algorithms to the input and training datasets are also relatively scarce.

It should be noted that labelled TC datasets exist for the past observed climate record (satellite data, reanalysis), but it may not be practical to generate such datasets in climate model outputs for every new simulation that is made and to which the detection algorithm is to be applied. Thus it is important to understand how a supervised method may depend on the training dataset if it is to be applied to a dataset of a slightly different nature. It is common practice that data from climate simulations are first produced and stored, and then analyzed. However, the climate modelling community is also moving in the direction of "on-the-fly" (also called in situ) data analysis in order to reduce the volume of data to be stored and the environmental impacts of such storage. This paradigm change implies the development of more efficient analysis methods. Both physical algorithms (i.e., trackers) and DL models are legitimate approaches to study TCs, but it is useful to understand if one generalizes better than the other. However, it is important to understand that both approaches do not necessarily achieve the same thing. Indeed, trackers search for the trajectory of a cyclone by detecting its different positions in time, whereas the DL models listed in Table 1, derived from computer vision, detect cyclones on an image frozen in time.

In this context and for the above-mentioned reasons, we have developed in this study a detailed procedure for building training datasets and testing the performance of the TC detection algorithm to some of its parameters. We chose to work with two reanalyses (ERA5 and MERRA-2) and view this as a necessary first step before being able to apply our methodology to climate simulations. In Section 1, we present the data used to generate the images to be classified. Then in Section 2, we explain the architecture of the classification model, its training, the evaluation method to assess its performances, as well as the processes for generating the images to be classified. In Section 3, we present the results of our experiments in terms of accuracy and the other evaluation metrics. We further present an investigation on misclassified images and some suggestions for future work. Finally we summarize our contribution in the last section.

## 2 Data

### 2.1 TC dataset

Several datasets of TCs exist: we can flag here ExtremeWeather (Racah et al., 2017), ClimateNet (Prabhat et al., 2020), and the International Best Track Archive for Climate Stewardship (IBTrACS, Knapp et al., 2010, and references therein). In this study we use the North Atlantic National Hurricane Centre (NHC) "best track" hurricane database (HURDAT2; available from www.nhc.noaa.gov/data/#hurdat; Landsea and Franklin, 2013) because it is known as a high quality dataset for the North Atlantic basin. Quality and quantity of the training dataset are essential for the accuracy and performance of ML models. In particular it is important for the dataset to be comprehensive (i.e., there is no missed TC) and homogeneous (i.e., the criteria for deciding if a feature qualifies as a TC are used consistently in space and time). The HURDAT2 dataset is reputed to be comprehensive for the period after 1970 (Landsea et al., 2010). It is more difficult however to ascertain its homogeneity especially for short duration TC.

HURDAT2 contains six-hourly (0, 6,12, 18 UTC) information on the location, maximum winds, central sea level pressure, and (since 2004) size of all known tropical cyclones and subtropical cyclones. The intensity of the TC are categorized into several categories, as shown in Table A1 in the Appendix. We consider the HU and TS categories as being TCs and the other categories (including tropical depressions) as not being TCs. We chose to exclude tropical depressions (corresponding to the HURDAT2 status TD) as well as the other weather events of lower intensity, because we wanted to focus on intense cyclonic events (> 34 knots) which are responsible for the largest impacts when they reach land.

## 2.2 Meteorological reanalyses

We use two different reanalyses upon which we train and apply our CNN. The ECMWF Reanaysis 5[th] generation (ERA5) is the current atmospheric reanalysis from the European Centre for Medium-Range Weather Forecasts (Hersbach et al., 2020). The Modern-Era Retrospective Analysis for Research and Applications, version 2 (MERRA-2) is the current atmospheric reanalysis produced by NASA Global Modeling and Assimilation Office (Gelaro et al., 2017). These two reanalyses differ in the atmospheric models used, the range of data being assimilated, and the details of the assimilation scheme. They also differ in their spatial resolution. ERA5 is retrieved from the ECMWF archive at a native resolution of $0.25° \times 0.25°$ while MERRA-2 is provided at a native resolution of $0.5° \times 0.6°$. The atmospheric variables relevant to TC detection are available in both reanalyses (as proposed by Liu et al., 2016). We use fields of sea level pressure, precipitable water vapor, the two components of the wind (at the surface and at 850 hPa) and the temperature at two different pressure levels (see Table 2). We have followed Liu et al. (2016) and considered an extensive set of meteorological variables to detect TC (see Table 2). This choice is confirmed by subsequent studies (Racah et al., 2017; Prabhat et al., 2020; Kumler-Bonfanti et al., 2020). It is likely that there is redundant information in this set of variables. An interesting follow-up work will be to investigate the relative contributions of these variables in the classification decision of the CNN, with the aim of reducing the number of variables.

## 2.3 Images

In computer vision, the term image refers to a stack of matrices (also called a 3D tensor), with each matrix representing an information channel. For example, RGB images are formed of a stack of matrices of numerical values coding the red (R), green (G) and blue (B) color intensities of each pixel of a photograph. Our use of the term image is a generalization of the concept of RGB images. In the rest of our study, an image refers to a stack of gridded data extracted from a different variable of ERA5 or MERRA-2 on a given geographical area. Unlike for an RGB image, the channels cannot be combined; we thus graphically represent each channel separately.

## 3 Methods

### 3.1 Classification model

In this study we implemented a binary classifier of cyclone images based on the work of Liu et al. (2016), with slight modifications. Table 3 shows the architecture of our CNN which is divided into two parts: a feature extraction part and a classification part. The feature extraction part is composed of the convolution layers whose filters are responsible for the extraction of features of cyclone present in the input images of the CNN. These features are the basic elements used for the classification of the images, implemented by the dense layers, and determine if the images represent cyclones or not, by outputting probabilities. As noted by Liu et al. (2016), using a shallow convolutional neural network is appropriate for a relatively small number of images in the training dataset because the network only has a small number of parameters to train.

Our modifications, compared to the work of Liu et al., concern the size of the convolutional filters and the number of neurons in the last dense layer. The characteristics of their CNN are described in Table 4. Indeed, our convolutional filters are smaller: 3×3 instead of 5×5 for Liu et al. We thought that smaller filters better capture the features of cyclones on small images, especially for the 16×16 pixels (px) images. In addition, 3×3 filters are more conventional now. Note that the number of trainable parameters are very much the same between our CNN and that of Liu et al.. Lastly, Liu et al. describe a final layer with two neurons using the logistic sigmoid activation function. This layer outputs two probabilities: the probability that the input image represents a TC and the probability that the image represents the background, but the outputs are not correlated and the sum of the probabilities can be larger than one. In this study, we use the conventional approach of binary classification by considering one output neuron activated by a sigmoid function. So a probability value that tends to zero classifies an image as background while a value that tends to one classifies an image as cyclone.

By construction, the size and the number of channels of the input images in a CNN are fixed. Using different image sizes and/or numbers of channels would require modifying the network architecture and retraining it. Indeed, the properties of the dense layers of the network depend on the image shape (i.e., the number of neurons). Thus, image classification using a CNN implies the production of training and testing datasets of a given shape, irrespectively of the atmospheric reanalyses, ERA5 or MERRA-2, being considered. In our study, the size of the images is 32×32 px or 16×16 px with the eight variables as the channels of the image (3D tensor). Of course, the channels must correspond to the same atmospheric fields in the same units across the two reanalyses and must be arranged in the same order. The next section explains how we tackled the production of such dataset of images.

### 3.2 Image preparation

#### 3.2.1 Principles

The training of a CNN classifier is based on the optimization of its parameters using gradient descent and backpropagation techniques. Roughly speaking, the training process presents a batch of images as an input to the CNN. The training process modifies the parameters of the CNN in order to improve the classification of the batch, according to a chosen loss function. For

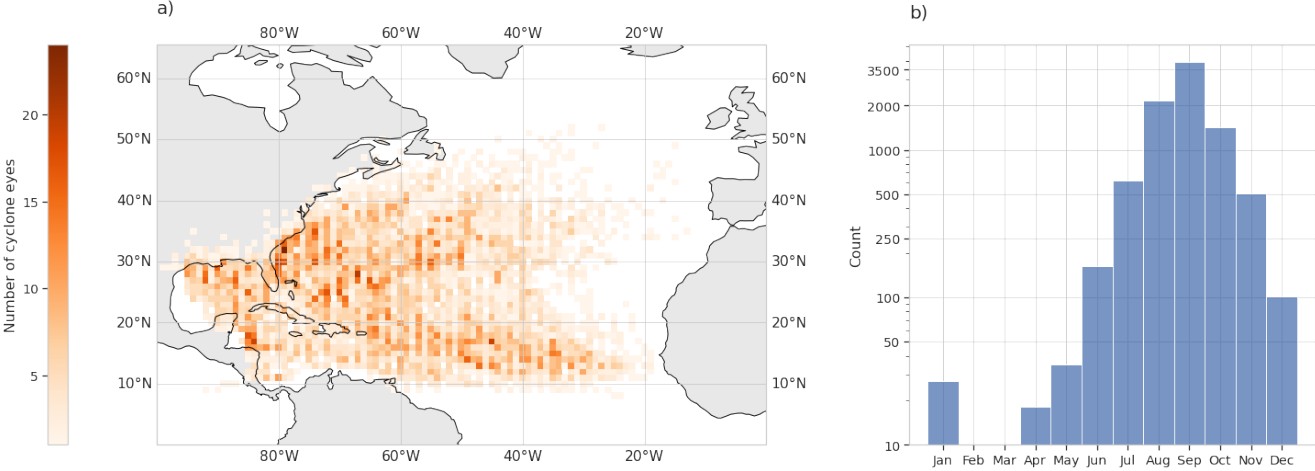

**Figure 1.** (a) Counts of TC-containing images per $1° \times 1°$ gridbox. (b) Histogram of the count of TC-containing images according to the month of the year. (a) and (b) are computed over the period 1980-2019.

a binary classifier, this process implies the presentation of images containing a TC, but also images not containing a TC, called background images. We now explain the data engineering involved in selecting both TC-containing and background images using the HURDAT2 dataset of cyclone tracks and the ERA5 and MERRA-2 reanalyses.

### 3.2.2 TC-containing image generator

The HURDAT2 dataset provides locations and dates of TCs as part of the cyclone metadata. We create images centered on the
155 cyclone positions in the reanalysis for the dates indicated in HURDAT2. The different channels of the images consist of the selected variables from the reanalyses as discussed above. We consider all cyclones with HU and TS status (see Table A1) that are located over the ocean, islands and coasts, over the period 1980-2019. Most TCs are found during the Atlantic hurricane season from May to December, but we also consider a few events identified by HURDAT2 as TCs outside these months. Figure 1 shows the spatial and temporal distributions of the TC-containing images.

### 3.2.3 Background image metadata generator

Extracting background images requires some thought because the performance of the CNN depends on these images and whether these images sample the diversity of TC-free situations. The idea here is to reuse the HURDAT2 database so that, for each location and date with a TC, we choose two dates in the past where no TC is present. We also check that the date was not already selected as the TC-free situation for another TC-containing image, so that all background images are distinct to each
165 other. Once the dates are selected, we can extract the corresponding images. Figure 2 shows a Unified Modeling Language version 2 (Object Management Group) activity diagram of the background image metadata generator and specifically how we compute the two dates from each date of a TC track. The first date is computed by subtracting between 48 and 168 hours

randomly (2-7 days) to the date of the TC track to generate the first date, and between 336 and 504 hours to generate the second date (2-3 weeks). Then the algorithm checks if each computed date leads to a background image that is in the immediate vicinity of any other TC track (status HU or TS as before) within a 48 hours time frame in the past or in the future, or to an already selected background image within a 12 hours time frame. If this is the case, we iterate by subtracting from the faulty date either 54 hours (48+time resolution) if the background metadata intersects a cyclone track or 18 hours (12 + time resolution) if it intersects another background image metadata.

Overall our background image metadata generator has the following advantages:

– our background images do not include a TC by construction;

– the meteorological, geographical and temporal contexts of the background images are close to those of the TC-containing images generated on the basis of the HURDAT2 data. In this way, we hope to better train the model at the classification decision boundaries;

– the ratio of background over TC-containing images is constant by construction (with one third TC-containing images and two thirds background images);

– the background images cannot be within 48 hours from a cyclone image and 12 hours from another background image, considering the geographical domain.

As a result of our image metadata generator, we obtain 9507 cyclone metadata and 19014 background metadata. The coordinates (longitude and latitude) of the cyclone and background metadata are then rounded to the respective resolutions of the ERA5 and MERRA-2 datasets, which results in two batches of metadata. Finally, we perform an additional step to check that no duplicate is created during the coordinate rounding.

### 3.2.4 NXTensor software library

The production of the image sets was the opportunity to create a reusable software library called NXTensor. This library is written in the Python 3.7 programming language and automates the extraction of geospatialized data, stored in NetCDF format, in a distributed and parallelized way on a computer cluster scheduled by Torque/Maui. Indeed, each channel of the images is produced by a task of the cluster (multitasking) and the extractions are performed in parallel (multiprocessing). The library ensures the determinism of the data extractions and it is reusable for other experiments than ours, because the parameters of the extractions are entirely configurable through yaml files. NXTensor takes as parameters the description files of the variables (path on the disk, naming conventions of the files, etc.), notably the period covered by the NetCDF files (e.g., ERA5 files are monthly while MERRA-2 files are daily), and the image metadata (date and location).

Figure 3 illustrates the step-by-step operation of NXTensor according to the UML2 activity diagram formalism, for the production of one of the channels of all the cyclone and background images. NXTensor starts by analyzing the image metadata to group them according to the period of the variable files to ensure that the files are only read once by distributed task. This analysis produces the block metadata, i.e. the set of data extractions to be performed by period. Then NXTensor submits

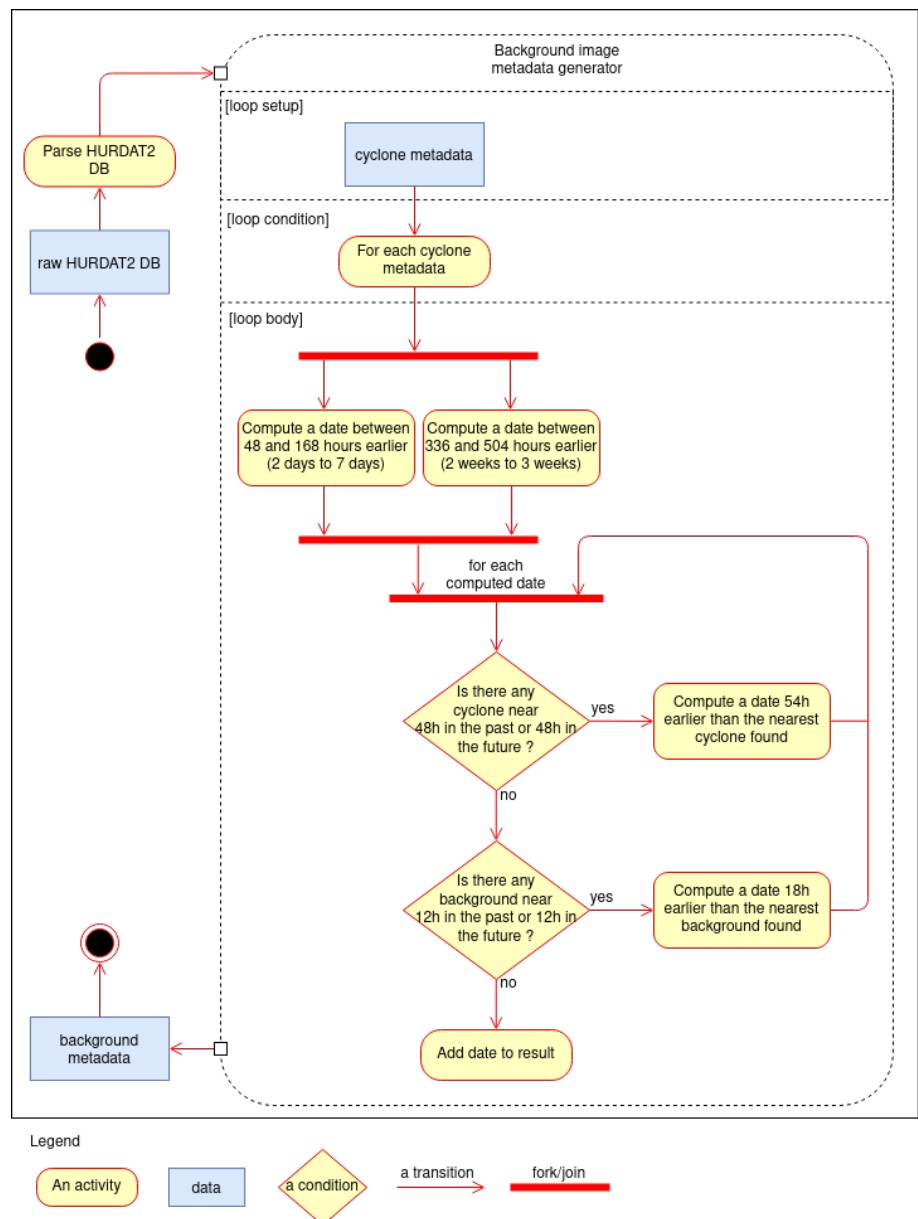

**Figure 2.** UML2 activity diagram of the background image metadata generator.

as many tasks to the cluster as there are channels, the determinism is ensured by sharing the same block metadata between the different distributed tasks. Within each task, the block metadata is divided into batches that are processed by a pool of workers performing the extractions of data in parallel. Each worker produces a set of blocks that are combined at the end by concatenation to form one of the channels of all the images. A special task is responsible for assembling the channels of the images in order to produce the 3D image tensor as mentioned above. For information, the elapsed time to extract a channel

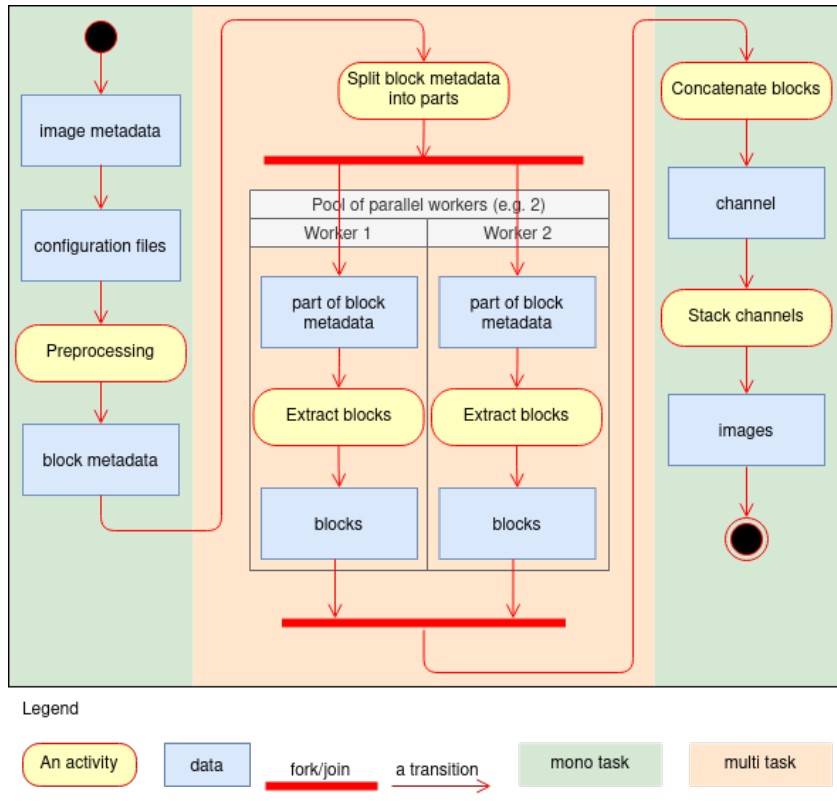

**Figure 3.** UML2 activity diagram of the image extractions using the NXTensor library.

for 28,521 images is about six minutes when the computations are carried out on the CPU cluster of the Institut Pierre-Simon Laplace (IPSL), using eight nodes (15 Go RAM and 15 cores AMD Opteron™ 6378 at 2.4 GHz). The channel assembly task takes about one minute. The CPU time was 135 minutes for the extraction of all the channels of the images.

### 3.2.5 Missing values issue

When generating images from the MERRA-2 data, we found that some of them had missing values (NaN), especially from the winds at 850 hPa. We decided to remove the metadata that resulted in incomplete images, for both the MERRA-2 and ERA5 batches, so that the batches of metadata are still identical. This resulted in the removal of 1,567 of them. Thus the number of cyclone metadata is 8,974 and the number of background metadata is 17,980, which gives a total of 26,954. With this final screening, we could then proceed to the extraction of the images.

### 3.2.6 Image interpolation

Previously we have detailed the automatic production chain of constant-shape images to satisfy the constraints of the CNN. However, as mentioned above, the ERA5 and MERRA-2 reanalyses do not have the same spatial resolution ($0.25°$ versus $0.5°$).

In order for the images to represent a constant domain size, and thus include cyclone of the same size as a fraction of the image domain size, we extract native images of 16×16 px for MERRA-2 and 32×32 px for ERA5 as described in Table 5. We then symmetrize the MERRA-2 native image set at a resolution of $0.5° \times 0.5°$ with a bi-linear interpolation to obtain the MERRA-2

16px@0.5 image set. To resolve the difference in resolution and to study the sensitivity of the CNN to the different datasets, we further transform by bi-linear interpolation one of the image sets to the properties of the other set (image resolution and size). Thus, we have two pairs of two image sets with similar properties: on the one hand ERA5 native and MERRA-2 32px@0.25 and on the other hand ERA5 16px@0.5 and MERRA-2 16px@0.5 (Table 5).

Figures 4 and 5 illustrate the representations of the channels of a cyclone and a background image, respectively, for the five

image sets, at a same localization, but for two different dates. It can be verified visually that the domain and pattern sizes of the images are independent of the choice of resolution. Finally, the input layer of the CNN is adapted dynamically to the size of the images during its instantiation, at the training phase which is described in Section 3.3.

### 3.2.7   Data standardization

Neural network models learn a mapping from input variables to output variables. The input variables have nearly always

different scales and large scale differences are detrimental to the learning process of neural networks. In order to ensure that each variable is equally important, regardless of its range of values, input variables are rescaled to the same scale. There are several methods such as standardization (or Z-score normalization) which consists in recalculating the values of the variables so that their mean and standard deviation equal to zero and one, respectively. In our study, we have systematically standardized each channel of the images, by calculating the means and standard deviations of the channels on all the images of the training

set. The validation and test image datasets are excluded from the calculation of the mean and standard deviation, to avoid that information about the validation and test datasets leak into the training phase. However, the validation and test datasets are also scaled using the mean and standard deviation of the training dataset.

### 3.3   Model training

We performed our model training experiments on HAL, a Dell GPU cluster available at the IPSL. Each of HAL computing

node is composed of two 2.6 Ghz Intel® Xeon® with four cores and two Nvidia® RTX® 2080 Ti 11 Go GPU cards, but only one card was used for our training experiments. On the software side, the model is implemented in Python 3.8, using the Keras 2.3.0 library which is a layer build on top of the Tensorflow 2.2.0 library, making it simpler to use. Overfitting has been noticed during the training of the model. We have observed the characteristic U-shape of underfitting followed by overfitting by plotting the value of the loss function calculated using the validation dataset against the number of epochs. In

order to automatically avoid overfitting, we used two Tensorflow callbacks: early stopping and model check point. The first callback stops the training after $N$ epochs without further improving the training metric ($N$ is set to a value of 10). Early stopping behaves more or less like the elbow method. The second callback always saves the weights of the model giving the best score of the training metric. As the number of epochs varies from one training to another (30 to 70), the training time also varies: between one and three minutes, knowing that one epoch takes less than one second of computation. Our work is

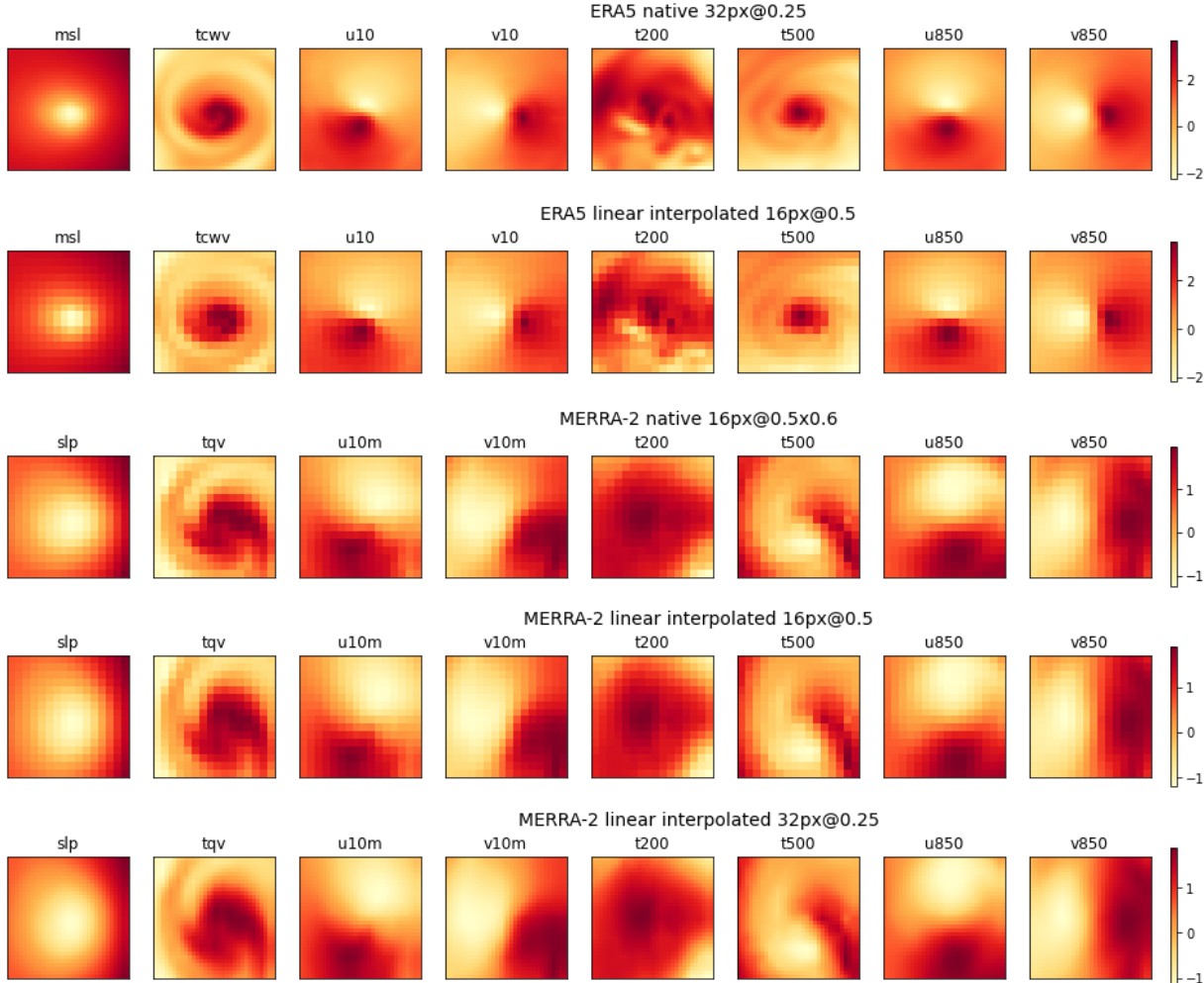

**Figure 4.** Channels (left to right) of the cyclone image on 22 August 1987, 00:00 UTC centered on 35.5°N, 43.125°W. The different rows show the native and interpolated images from ERA5 and MERRA-2 as per the labels. Data is standardized.

based on the study by Liu et al., but these authors did not provide the values of their training hyperparameters such as batch size, optimizer and learning rate. Instead of fixing these values in an arbitrary way, we search for local optimal values of these hyperparameters to maximize the performance of the CNN. Since training times are relatively short on our GPU cluster, we performed a grid search hyperparameter optimization to maximize the score of the training metric, using conventional hyperparameter value ranges (the number of combinations of the search space is 48). We conducted four optimizations for the different image datasets, but for the same training / validation / testing split (0.70/0.15/0.15). We obtained the same values for the optimizer and the learning rate, with very close performances. Only the batch size differs, so we decided to set a value as large as possible given the memory of the GPU cards at our disposal. Of course, these optimal values are only valid for the given split, however we think that they are close to the global optimum, because the performances vary very little according to

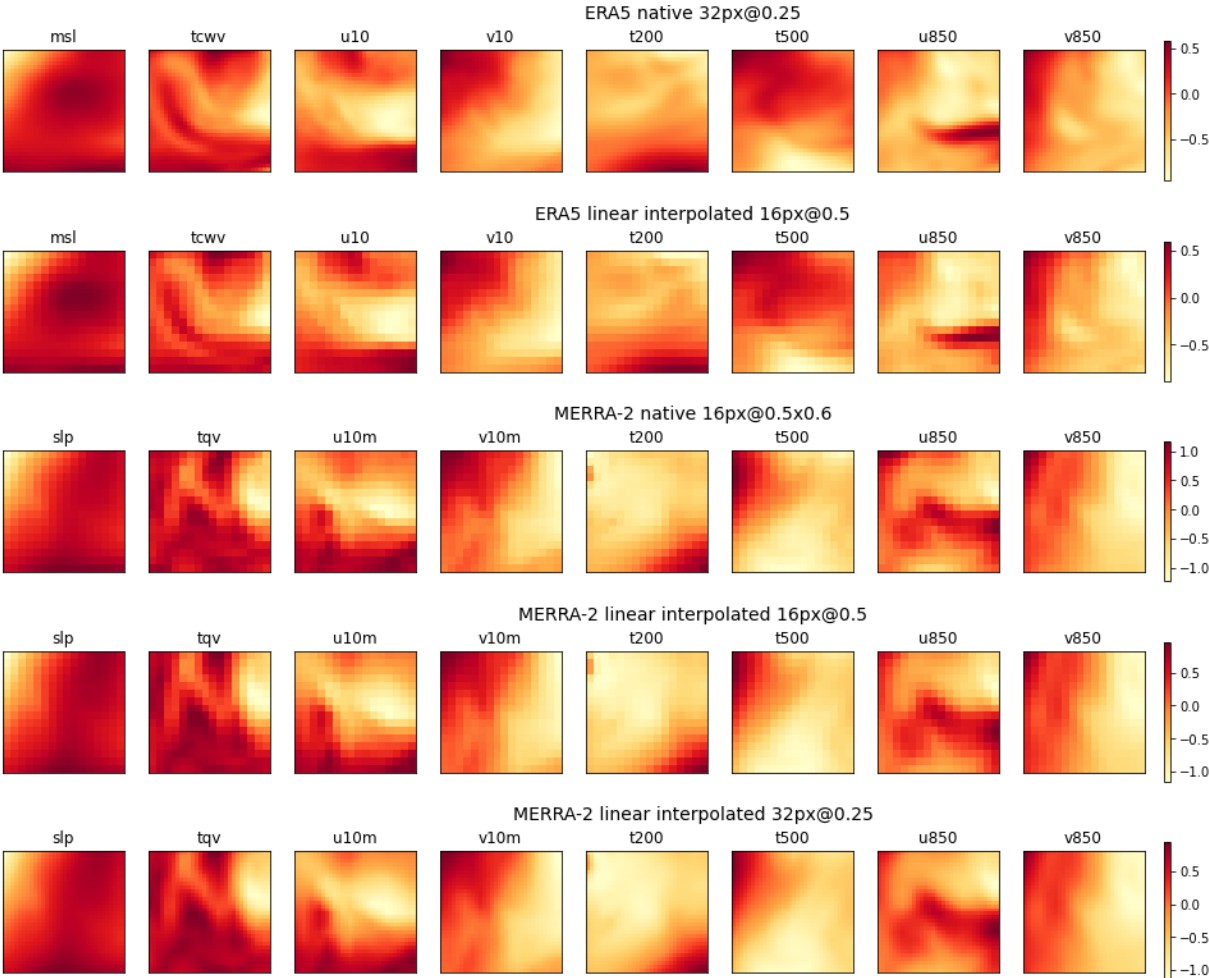

**Figure 5.** Same as Fig. 4, but for the background image on 6 August 1987, 18:00 UTC centered on 35.5°N, 43.125°W. Data is standardized.

the different values of these hyperparameters. The obtained values, described in Table 6, are used for all experiments to avoid
attributing the variability of the studied metrics to hyperparameter changes. These metrics and the methods for evaluating them
are the subject of the next section. We did not try to optimize the architecture of the CNN proposed by Liu et al. because we
believe they have already optimized it and the modifications we have made do not require any further optimization, since the
performance of our CNN is very close to that of Liu et al. Finally, as explained above, we prefer to focus on the performance
and the sensitivity of the CNN to the learning dataset instead of obtaining better performances.

## 3.4 Evaluation of metrics

In our study, we used three classical metrics to measure the performance of our binary classification model: accuracy, the
Area Under the Curve (AUC) of the Receiver Operating Characteristic (ROC) and the Area Under the Precision-Recall Curve

(AUPRC). The equations of the binary classification metrics are given in Appendix C1. The accuracy measures the rate of good predictions of a model. It is an easy metric to interpret, but it depends on the decision threshold for which the value of a probability is associated with one class rather than the other. It was criticized in particular by Provost et al. (1997) and Ling et al. (2003) and we discuss it further in Section 4.1. The AUC measures the power of a model to discriminate between the two classes for a variety of decision threshold values. The AUC is, as the acronym indicates, the area under the ROC curve which is plotted by the values that recall (the ability of a model to identify all occurrences of a class) and false positive rate of a model take (definitions are given in Appendix B1). The recall and false positive rate values are calculated according to the ground truth and the classifier responses for a given test dataset and for all possible decision thresholds (or a set of discrete values). A perfect classifier has an AUC equal to 1, recall all the images of cyclone with a null false positive ratio. The AUPRC measures the recall of a model while minimizing the precision (prediction errors). The AUPRC follows a similar approach to the AUC: it is the area under the curve which is plotted by the values that the precision and recall of a model take (for all possible decision thresholds, etc.). A perfect classifier has an AUPRC equal to 1, that is, recall all the images of cyclone without wrongly classifying any background image as a cyclone image. AUC and AUPRC are much more interesting because they are integrated on the decision threshold values.

For the evaluation and comparison of the metrics (developed in Section 4.2), we wanted to be able to calculate the expected value and the uncertainty of the metrics, without bias. To that end, we applied an iterative cross validation method which consists in repeating 20 times a cross validation method. We chose the $k$-fold method (Bishop, 2006), with $k$ equal to ten, as the cross validation method. We obtained a mean of the metrics for each $k$-fold iteration. By applying the central limit theorem on this set of metric means, we could compute the expected value and the uncertainty of the metrics.

In order to avoid any bias, we took care to check if the central limit theorem can be applied, by testing the normality of the distribution of the metric means using the Shapiro-Wilk statistical test (brief non-mathematical presentation given in Appendix B1). Moreover, images coming from a time series of tracks from the same cyclone may be found in both the training and test datasets, which would induce some dependance between the training and test datasets due to the autocorrelation within individual cyclone tracks. In order to avoid such a bias, the $k$-fold split is based on sampling the years randomly and balancing the folds as much as possible. The partitioning combinations are calculated in advance in order to guarantee the uniqueness of their composition. Scale bias is also avoided by standardizing the channels of the images online, just before training the CNN.

Finally, for the comparison of the metric means, we chose to apply the Kruskal-Wallis statistical test (brief non-mathematical presentation given in Appendix B2) for an alpha level of 1 %, because the Shapiro-Wilk test was negative for most distributions of metric values of our experiments, invalidating the use of the Student's $t$-test.

For the experiment of highlighting the problem with the accuracy (point developed in Section 4.1), we applied the classical hold-out method, avoiding the autocorrelation between images belonging to a same cyclone track, with the following partitioning: 70 % of the data for the training dataset and 30 % of the data for the test dataset.

# 4   Results

## 4.1   Accuracy and its threshold

Accuracy is a convenient measure, but according to Provost et al. (1997) and Ling et al. (2003), the class threshold makes it non objective. Provost et al. (1997) argue that in real world cases, the use of accuracy as an ML model metric is questionable at best, because the distribution of classes is generally not known so it is impossible to optimize the misclassified rate. Furthermore, the authors demonstrate while in some examples the classifiers have their accuracy statistically comparable, their AUC is significantly different. Ling et al. (2003) argue that the accuracy loses information during the transformation of the probability, returned by the classifier, into a class identifier: as soon as this probability exceeds the class decision threshold, the response of the classifier takes the class identifier while the information of the difference between the value of the probability and the threshold is lost. In order to provide further evidence of this problem, we study the distribution of the classifier's predictions using the hold-out method. Rather than applying it on a single set of images, we identically partitioned the four sets of images and trained and tested the classifier for all possible combinations. Figure 6 shows plots of the distributions as log-scale histograms, colored according to the ground truth of the images. Then we calculated the decision threshold for which the number of misclassified predictions is minimal. For this purpose, we use the Youden's index which is a measure of the tradeoff between sensitivity and specificity. Maximizing the index means minimizing the false positives and false negatives according to its equation described in Appendix C6, knowing that Youden's index varies according to the class decision threshold. By default, ML libraries set the threshold to 0.5; however, in our case, the optimal threshold, indicated in the title of each subplot of Fig. 6, is lower than 0.5, and for some combination of training and testing datasets, even much lower (e.g., ERA5/MERRA-2 combination in 16px@0.5). This reflects the fact that i) the image sets are not balanced and ii) the number of false negatives (orange color on the left side) is larger than the number of false positives (blue color on the right side) for this particular partitioning.

Our set of experiments shows that the choice of the threshold value depends on the partitioning, the source of the data and the relative importance given to false negatives and false positives. While accuracy is a less interesting metric than AUC and AUPRC, we decided to keep it, as a matter of information, and set its threshold to 0.5.

## 4.2   Metric comparisons

### 4.2.1   Inter-comparisons

In this section, we focus on the values of the CNN metrics obtained using the iterative cross-validation method on each of the image sets described in Table 5. Since the Shapiro-Wilk test shows that the distribution of the iteration means is normal for all metrics, under the central limit theorem, we computed the expectation and standard deviation of each of the metrics, given in Table A2. The values of the metrics are very high, over 0.98. They are close to those reported in the studies about image classification that we have listed in Table 1. As a comparison, the accuracy value of our CNN is between those of Shakya et al. (2020) and Liu et al. (2016): $0.97 < 0.98 < 0.99$. However, this comparison should be put into perspective: our method

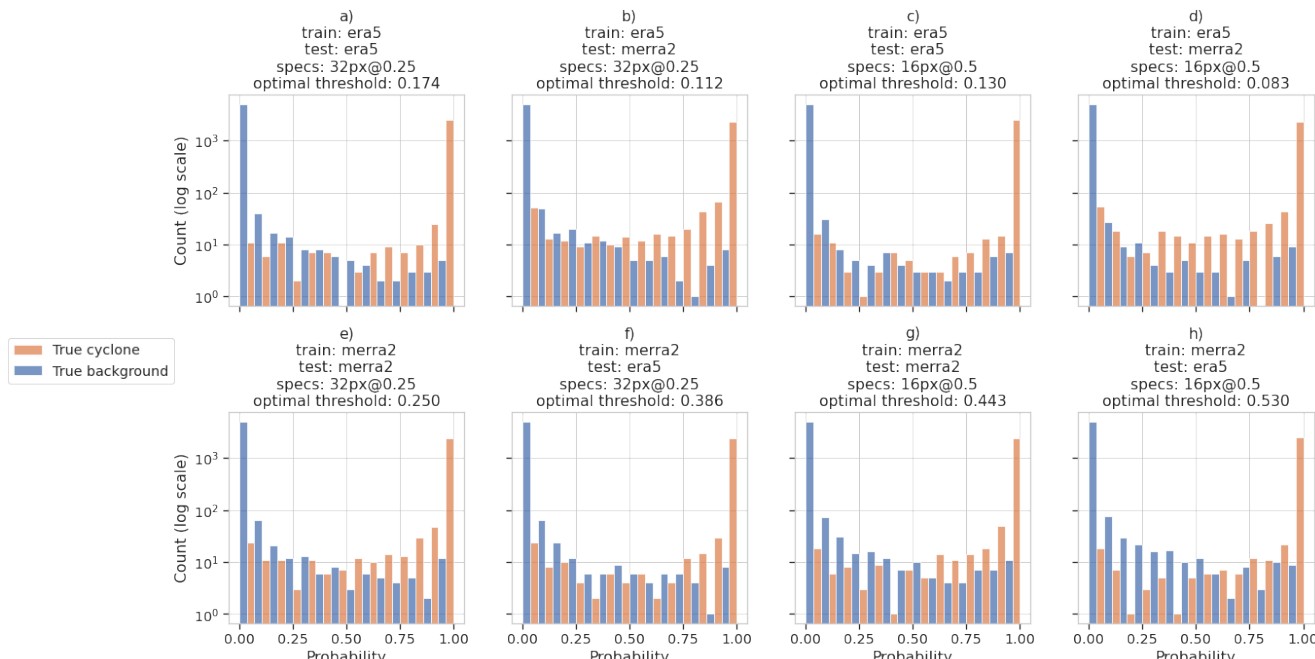

**Figure 6.** Histograms of the predicted probabilities for "true cyclone" images (orange bars) and background images (blue bars) for different combinations of training and test datasets and resolution. The optimal decision threshold is indicated in the title of each histogram. Note the logarithmic scale on the $y$-axis and that by construction there is twice as many background than cyclone images.

of calculating the accuracy is more robust against uncertainty. Additionally, the model of Shakya et al. (2020) is trained and tested on observational data that are quite different from multidimensional meteorological reanalysis data.

The values of the metrics are very high, but it does not mean that a model is useful. Indeed, the usefulness of a model is measured by the difference between its performance and that of models based on simple rules or a domain specific baseline. For instance, we implement the following simple models (from the software library scikit-learn): "most frequent" which always predict the most frequent class observed in the training dataset (i.e., background), "stratified" which generates randomly predictions at probabilities that respect the class distribution of the training dataset (i.e., 1/3 cyclone, 2/3 background), "uniform" which generates predictions uniformly at random background or cyclone with equal probability. In our study, the CNN

performs significantly better than simple models as shown in Fig. 7. By plotting the values of the metrics on Fig. 8, we can see that although very close, the performances of the CNN are grouped according to their original dataset (MERRA-2 and ERA5) and that the performances of these two groups seem significantly different. In order to have an objective confirmation, we chose to compare the values of the metrics using the Kruskal-Wallis test, as the distributions of the metric values are not mostly normal (see Table A2). Table A3 summarizes the pairwise comparison of the metric performances according to the image set

used and confirms our interpretation of Fig. 8. This experiment tells us that the difference between the metric values computed from the same dataset, interpolated and not interpolated, can be attributed to randomness. Whereas the metrics computed from

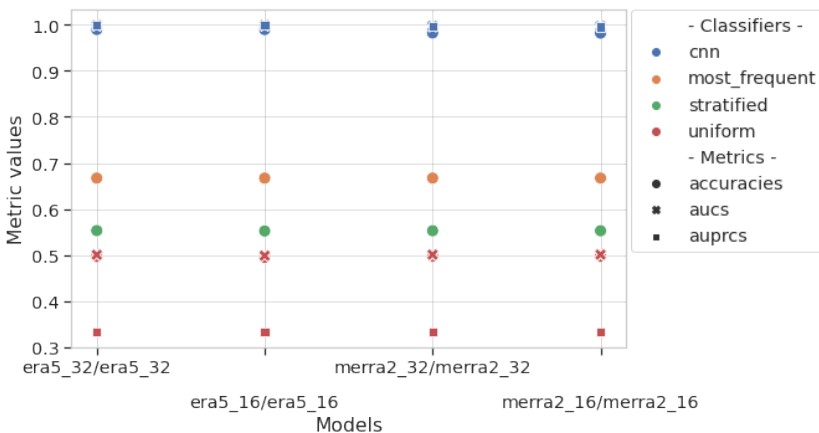

**Figure 7.** Metric values showing the performances of the CNN versus simple classifiers for the four datasets. The color of the symbols corresponds to the classifiers while the marker shapes indicate the nature of the metric. The models are tested against the same image set as they are trained against (e.g., era5_32/era5_32 means the CNN was trained and tested on ERA5 native).

different dataset are quite distinct. So in our study, we can say that the interpolation does not impact the model performance and training with interpolated datasets has some meaning. At last, we observe that the values of the metrics from ERA5 are greater than those from MERRA-2.

**4.2.2   Cross-comparisons**

In this section we are interested in the values of the metrics of the CNN trained on one image set and tested on the other image set with the same properties (image resolution and size). In the same way as the previous experiment, we computed the expectation and standard deviation for each of the metrics, given in Table A4, and then compared the performance obtained previously (training and testing with the same image set) with these values (training and testing with a different image set).
Figure 9 gives the graphical representation and Table A5 gives the result of the Kruskal-Wallis tests. This experiment shows us that regardless of the resolution and the dataset used for model training, the metric values are statistically distinct and the value of the metrics evaluated on the ERA5 dataset is greater than that evaluated on the MERRA-2 dataset. Thus we can conclude that the ERA5 dataset is more information rich than the MERRA-2 dataset for the classification of cyclone images using our CNN.

**4.3   Misclassified images**

Following the comparison of the metrics, we took a closer look at the metadata of the images misclassified by the CNN. Table A6 in the Appendix summarizes the number of false alarms for each combination of training and testing datasets discussed in Section 4.1. We studied the metadata of the failed predictions that are common to all training/testing datasets so as to limit the study to the most significant cases. We also contextualize the misclassified images in the HURDAT2 time series. There is

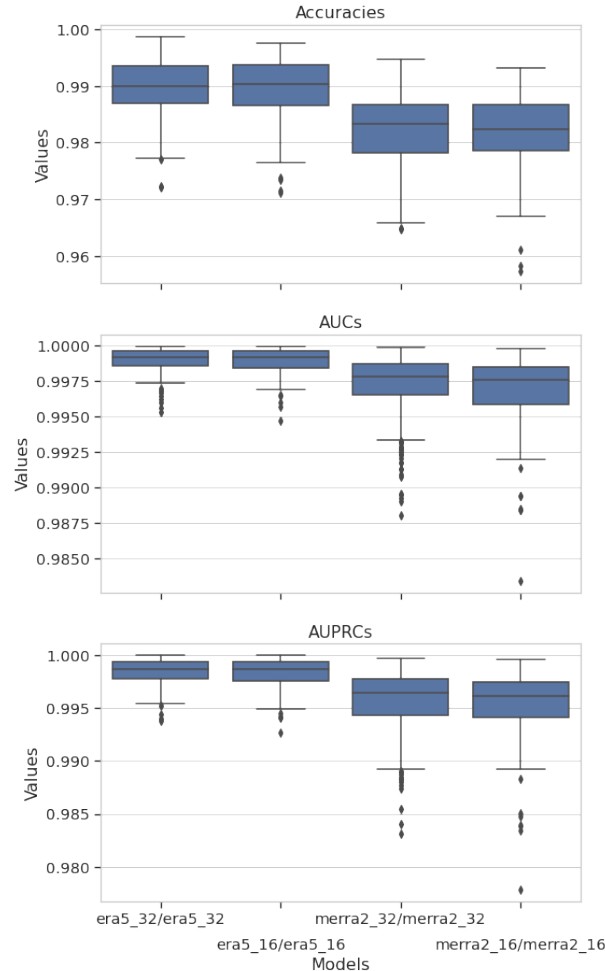

**Figure 8.** Box plots of the accuracy, the AUC and the AUPRC metric values for the CNN for the four image sets. The models are tested against the same image set as they are trained against (e.g., era5_32/era5_32 means the CNN was trained and tested on ERA5 native). Box plots are a synthetic representation of a data distribution. They are composed of a box and two whiskers. The bottom of the box corresponds to the first quartile ($Q_1$) of the studied dataset, below which 25 % of the data are located. The middle line of the box is the median of the dataset, and the top line of the box corresponds to the third quartile ($Q_3$), below which 75 % of the data are located. The interquartile range is represented by the extent of the box: $IRQ = Q_3 - Q_1$. The bottom whisker is calculated according to the $IRQ$: $Q_1 - 1.5 \times IRQ$ and the top whisker: $Q_3 + 1.5 \times IRQ$. The data outside the whiskers are considered as outliers, and represented as diamond markers.

a total of 15 false alarms in common, i.e. seven false positives (background images wrongly classified as cyclones) and eight false negatives (cyclone images wrongly classified as background). However, we found that the false negatives and the false positives were generated from the tracks of the same cyclones. Thus, after removing the duplicates, there are only eight false alarms left in common, i.e. seven false positives and one false negative.

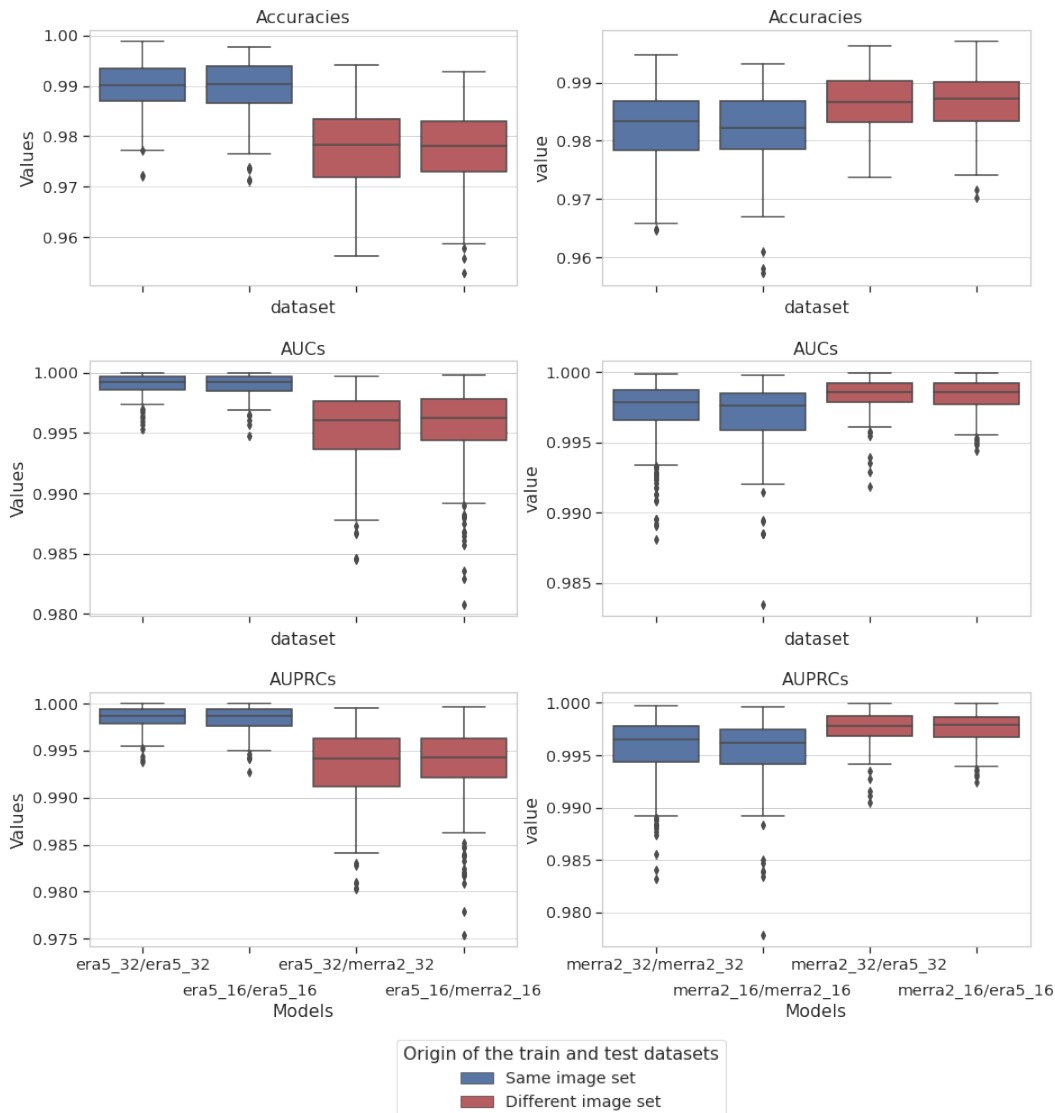

**Figure 9.** Box plots of the accuracy, the AUC and the AUPRC metric values for the CNN for different combinations of training and test image sets. In blue, the models are tested against the same image set that they are trained against. In red, the models are tested against the other image set of the same resolution (e.g., era5_32/merra2_32 means the CNN was trained on ERA5 native and tested against MERRA-2 32px@0.25).

### 4.3.1 False Positives

First, we studied the false positives and have listed them in Table A7. For each image, we gave their HURDAT2 status (see Table A1) as well as the average probability given by the CNN for each dataset (mean prob column). For each of the false

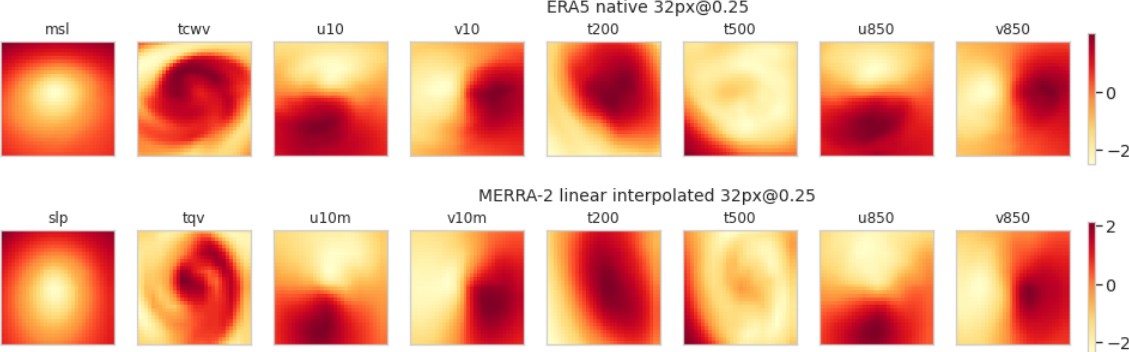

**Figure 10.** Channels of the image on 5 August 1990, 00:00 UTC centered on 38°N, 30.625°W, taken as an example of a wrongly classified image as a cyclone (false positive). Data is standardized.

positives, we verified if there was a cyclone close in the past and in the future, by querying the HURDAT2 database and indicated the number of hours that separate them from a referenced cyclone (status HU or TS; respectively the past and future columns). What we can already observe is the high value of the mean probability and its low standard deviation: the CNN is
wrong with high confidence for these images whatever the dataset used, which confirms the relevance of the failed predictions in common. Then, we notice that these images are temporally close to a TC by an average of 131 hours, approximately five days and a half (in the past or in the future). Thus we deduced that the false positives are essentially linked to transition states leading to a cyclone or to its dissipation. Figure 10 gives a graphical example of one of these false positives for the ERA5 and MERRA-2 image sets.

### 4.3.2  False negative

We have list the single example of false negative that the training/testing datasets have in common, in Table A8 and we give a graphical example in Fig. 11. The image refers to a cyclone which status is TS and we give its mean probability and standard deviation computed by the CNN for each dataset. We computed the lifetime of cyclonic activity near the geographical area of this image, as previously by querying the HURDAT2 database and indicated the number of hours that separate this image from
the first track of a cyclone in the area. We observe that the probability is very low that means that the CNN is wrong with high confidence and the low standard deviation of this probability means that this false negative classification is relevant for all the combination of training/testing datasets. We also notice that this image is temporally close to a tropical depression, six hours in the future, suggesting that this false negative is essentially linked to the dissipation of a stationary cyclone.

## 5  Discussion and potential future work

Designing, optimizing and testing a DL method for image classification involves many modeling choices, some of which we have assessed, some of which we have not. We now discuss some of the choices we have made and potential future work.

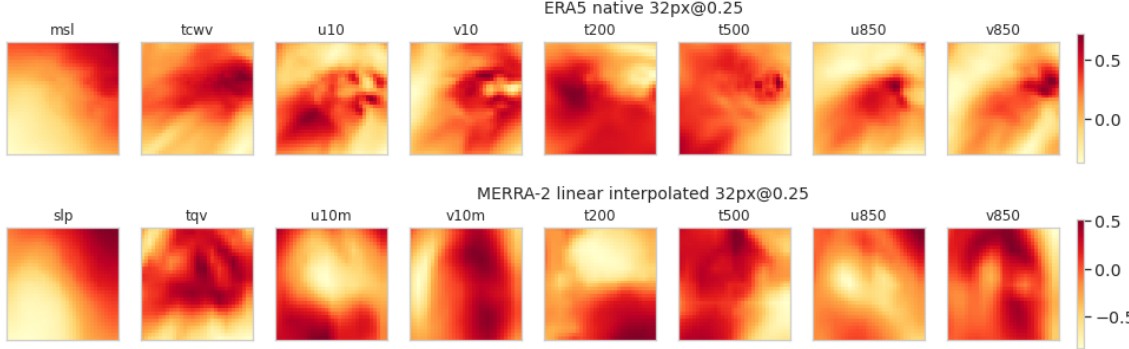

**Figure 11.** Channels of the image on 6 August 1990 06:00 UTC, centered on 27°N, 46.875°W, taken as an example of a wrongly classified image as background (false negative). Data is standardized.

We have chosen a binary approach for the classification (i.e., TC or background), but it is quite possible to design a classifier predicting the range of HURDAT2 status of the images. Such a classifier would use nine neurons with the soft max activation function as the last layer of the CNN. However, training it would probably face an acute problem of image set imbalance. Indeed, four classes out of nine have a number of occurrences smaller than 400 in HURDAT2 (see Fig. A1). To improve the situation, it would be possible to merge some classes (e.g., WV with DB and SD with SS) in order to mitigate the problem.

Although the performance of our CNN is not an issue (AUC and AUPRC are over 0.99), it may be less satisfactory in other settings or if applied to classify other meteorological features. Several leads can be pursued to improve the performance in the future.

Our intercomparison experiments have shown that a bi-linear interpolation does not affect the performances of the classifier. However, there are other interpolation methods like bicubic, nearest neighbor, etc. It would be interesting to verify whether these interpolation methods have any effect on the performance of the classifier. The choice of interpolation method is particularly relevant in the case of an intercomparison of data from multiple sources with different native resolutions, such as is the case in climate model intercomparison studies.

Our cross-comparison experiments have shown that applying an ERA5-trained CNN on MERRA-2 images works better than applying a MERRA-2 trained CNN on ERA5 images, which suggests that ERA5 has a larger information content in the framework of our CNN. This is also consistent with the findings of Malakar et al. (2020) who analyzed the error in the location of the center, maximum winds and minimum pressure at sea level in six meteorological reanalyses including ERA5 and MERRA-2 for the evolution of 28 TCs occurring between 2006 and 2015 over the North Indian Ocean, with respect to the observations of the Indian Meteorological Department (IMD). The authors of this study show, among other things, that the ERA5 dataset captures the evolution of these TCs in a more realistic way than MERRA-2 (i.e., smaller errors in the previous variables). They also show that ERA5 and MERRA-2 can capture the intensity of the TCs from the depression stage to the very severe cyclonic storm stage, but not from the extremely severe cyclonic storm stage for which the intensity of the TCs is underestimated. However, they conclude that of the six datasets, ERA5 provides the best representation of the TC structure in

terms of intensity. Finally, the study published by Hodges et al. (2017) shows that 95 % of the Northern Hemisphere TC tracks, from the IBTrACS database that includes HURDAT2, are present in MERRA-2. Unfortunately, this study does not include ERA5. It also confirms the underestimation of cyclone intensity in MERRA-2 compared to observations.

Some transfer learning experiments would also be interesting to conduct. For example, instead of training the CNN with randomly initialized weight values, training the CNN on one image set with weight values initialized with those of the CNN trained on the other image set with the same properties could improve the performance of the CNN.

Data augmentation (especially geometric transformations; Shorten and Khoshgoftaar, 2019) and model regularization techniques (e.g., weight-decay, batch normalization, dropout, etc.) are proven ways to improve the robustness of a CNN trained with a dataset of limited size. Our dataset contains 26,954 images, which is relatively small compared to the size of datasets encountered in many computer vision applications (for instance, Imagenet contains more than 14 million images). However, using these techniques was not justified for our study because the performance of our CNN without data augmentation is already very high (AUC and AUPRC are over 0.99). Such techniques could however become very relevant in future work when we seek to detect TC in climate model simulations with a CNN trained on a reanalysis dataset. Indeed different climate models may simulate TC imperfectly and there is probably some value in offering a larger variety of TC structures to the training dataset. It is expected that the simulation of TCs increases in quality with the climate model resolution (Strachan et al., 2013) and climate models running at resolutions of 10 to 50 km are now commonplace. Likewise we would need to augment the number of images with very intense TC or TC migrating outside their usual domains because there are indications that such situations may become more frequent with global warming (as presented in the introduction) and we want to ensure these can be detected adequately in climate simulations.

In this study we work on images created on a regular lat-lon grid, which potentially introduces a deformation because of the $\cos(\text{latitude})$ dependence of a displacement element along the longitude. Such a deformation is small in the tropical region and therefore is not thought to be a problem for our analysis. However, it increases as a function of latitude, so it may become an important factor to consider for TC that migrate polewards or for the detection of mid-latitude depressions. Data augmentation techniques that introduce deformed images in the training datasets could help to increase the robustness of the CNN in these situations.

Finally, pixel attribution experiments (saliency maps) should give us the importance of each variable, with hints on a possible reduction of their number or on the use of composite variables such as vorticity. These experiments could also give explanations on misclassified images. Occlusion-perturbation based methods like local surrogate (LIME; Ribeiro et al. 2016), Shapley values (SHAP; Lundberg and Lee 2017), and gradient based methods like Grad-CAM (Selvaraju et al., 2017) should be resourceful.

## 6   Conclusions

In this study, we have adapted and tested a CNN for the classification of images according to the presence or absence of tropical cyclones. The image sets for training and tests were built from the ERA5 and MERRA-2 reanalyses with labels derived from the HURDAT2 dataset. We have paid a lot of attention on the design of the background image set to make sure it

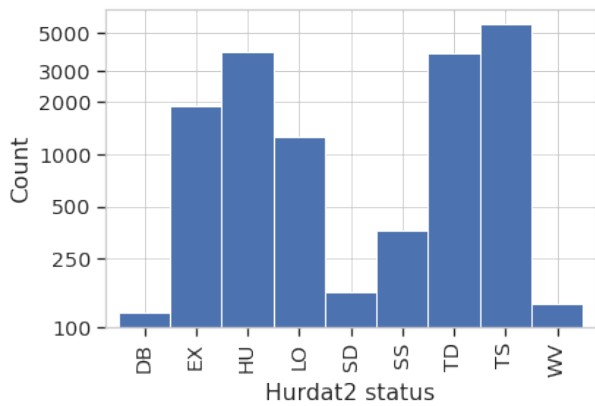

**Figure A1.** Distribution of the cyclone categories/status computed over the period 1980-2019

samples similar location and time to the TC-containing image. We have assessed the performance of the CNN using accuracy, but also the more objective AUC and AUPRC metrics. We have shown that failed classifications may be explained by the meteorological context. In particular false positives often represent a situation with cyclonic activity, but not yet classified as TC by HURDAT2. It should be relatively easy to diagnose those situations if the TC are tracked in time rather than dealt with as a set of separate independent images as it is the case in this study. We have further shown that interpolation (from $0.5°$ to $0.25°$ or from $0.25°$ to $0.5°$) does not impact the performance of the CNN and an ERA5-trained CNN on MERRA-2 images works better than applying a MERRA-2 trained CNN on ERA5 images. This study paves the way for a future study aiming to assess the performance of an automatic detection scheme of TC in climate simulations without a specific retraining of the CNN for each new climate model or climate model resolution.

*Code and data availability.* The image sets computed from ERA5 and MERRA-2, their metadata, HURDAT2 data and the code used in this work (experiments and NXTensor) are all available at this address: https://doi.org/10.5281/zenodo.6881020 (DOI: 10.5281/zenodo.6881020). The code is open source and distributed under the CeCILL-2.1 license. More information about HURDAT2, ERA5 and MERRA-2, including how to download them, is available from https://www.nhc.noaa.gov/data/#hurdat, https://www.ecmwf.int/en/forecasts/datasets/reanalysis-datasets/era5, https://gmao.gsfc.nasa.gov/reanalysis/MERRA-2/, respectively.

## Appendix A: Tables

## Appendix B: Statistical tests

Let us consider a group, a set of values of a random variable that we observe, for example during an experiment, and its population, the set of all the values that the variable can take for a particular experimental context. The group is a subset of the population.

## B1  Shapiro-Wilks test

The Shapiro-Wilks test poses the so-called null hypothesis (H0) that the group of values that a given quantitative variable takes comes from a normally distributed population. For a non-technical explanation of the test, we can say that the Shapiro-Wilks test quantifies, using a single metric (the p-value), the dissimilarities between the distribution of the values of the group and the distribution of the population of the variable if it was normal. For a risk of error called alpha level, commonly fixed at 1 % or 5 %, H0 null is rejected if the p value is lower than the alpha level and is accepted if the p value is higher than the alpha level. The latter represents the risk of accepting H0 when it is not true: a false positive. If H0 is accepted, random sampling of the group can explain the dissimilarities between the distribution of the values of the group with a normal population distribution. If H0 is rejected, it can be stated that the population of the variable is not normally distributed.

## B2  Kruskal-Wallis test

The comparison of the mean of the values of a variable from two different groups is usually done by the Student's t-test (or two-sample ANOVA) or the Kruskal-Wallis test. The Student's t-test is a parametric statistical test, in this case it requires that the distribution of the given variable is normal. The Kruskal-Wallis test is non-parametric and does not require the assumption of the population distribution of the variable. Indeed, it is not based on the value that the variable takes, but on its rank in the classification of the observed values of the variable. As for the Shapiro-Wilks test, we propose a non-technical explanation of the Kruskal-Wallis test: this test quantifies, using a single metric (the p-value), the dissimilarities between the mean ranks computed for two or more groups of values, with, as H0, random sampling of the group being able to explain the differences between the medians of the two groups, because they may come from the same population.

## Appendix C:  Equations

TP, TN, FP, FN stand for True Positives, True Negatives, False Positives, and False Negatives, respectively.

## C1  Binary classification metrics

The binary classification metrics used in this study are defined as:

$$\text{Accuracy} = \frac{TP + TN}{TP + TN + FP + FN} \tag{C1}$$

$$\text{Precision} = \frac{TP}{TP + FP} \tag{C2}$$

$$\text{Recall or Sensitivity} = \frac{TP}{TP + FN} \tag{C3}$$

$$\text{False positive rate} = \frac{FP}{FP+TN} \tag{C4}$$

$$\text{Specificity} = \frac{TN}{TN+FP} \tag{C5}$$

## C2 Youden's index

The Youden's index is defined as:

$$J = \text{sensitivity} + \text{specificity} - 1 = \frac{TP}{TP+FN} + \frac{TN}{TN+FP} - 1 \tag{C6}$$

*Author contributions.* SG designed the study, including the data selection and cross-validation evaluation procedures, developed all codes used in this study, carried out all experiments, and performed the analysis. OB contributed to the design and interpretation of the results. SG and OB both contributed to the writing of the manuscript.

*Competing interests.* The authors declare no competing interests.

*Acknowledgements.* This study benefited from the ESPRI computing and data centre (https://mesocentre.ipsl.fr) which is supported by CNRS, Sorbonne Université, Ecole Polytechnique and CNES as well as through national and international grants. The authors acknowledge funding from the French state aid managed by the ANR under the "Investissements d'avenir" programme with the reference ANR-11-IDEX-0004-17-EURE-0006 and from the European Union's Horizon 2020 research and innovation program under grant agreement 101003469 for the "eXtreme events: Artificial Intelligence for Detection and Attribution" (XAIDA) project.

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

**Table 1.** Summary of previous studies aiming at the detection or the segmentation of TC in satellite or model data. coeff: coefficient; IoU: intersection over union; IR: infrared; mAP: mean average precision; mod: model; rea: reanalysis; RMSE: root mean square error; sat: satellite; vis: visible.

| Authors | Purpose | Dataset | Variables | NN architecture | Performance at best |
|---|---|---|---|---|---|
| Liu et al. (2016) | Cyclone image classification | ERA-interim (rea), CAM5.1 (mod), NCEP-NCAR (rea), 20th Century Reanalysis | Pressure sea level, wind vectors at 10 meters and at 850 hPa, temperature at 200 and 500 hPa, total column water vapour | ad hoc CNN | Accuracy: 99 % |
| Racah et al. (2017) | Cyclone segmentation | CAM5 (mod: 25 km) | The 16 channels of CMA5 | ad hoc autoencoder | mAP@IoU=0.1: 52.92 % |
| Hong et al. (2017) | Eye detection | COMS-1 (sat) | 4 IR channels | GoogLeNet | RMSE: 0.02 |
| Matsuoka et al. (2018) | Detection of cyclone by sliding window and cyclone image classifier | NICAM (mod: 14 km) + NSW6 + HadISST | Outgoing longwave radiation | ad hoc CNN | Probability of detection: 79.9–89.1 %, false alarm ratio: 32.8–53.4 % |
| Kumler-Bonfanti et al. (2020) | Cyclone segmentation | GFS (mod) | Total precipitable water | U-Net | Accuracy: 0.991 %, Dice coeff: 0.763, Tversky coeff: 0.75 |
| Shakya et al. (2020) | Cyclone image classification | KALPANA-I MOSDAC (sat) | IR, Vis | ad hoc CNN | Accuracy: 97 % |
| Shakya et al. (2020) | Cyclone detection and path prediction | KALPANA-I MOSDAC (sat) | IR, Vis | RetinaNet and polynomial regression | RMSE: 5-15.55 % |
| Prabhat et al. (2020) | Cyclone segmentation | CAM5.1 (mod: 25 km) | Integrated vapor transport, integrated water vapor, vorticity, wind vectors at 10 meters and at 850 hPa, sea level pressure | DeepLabv3+ | IoU: 0.2441 |
| Pang et al. (2021) | Cyclone detection | Satellite images from NII | Vis | DCGAN and YOLOv3 | Accuracy: 97.78 %, mAP@IoU=0.5: 81.39 % |
| Shi et al. (2022) | Extratropical cyclone detection | ERA5 (rea) | Top net thermal radiation, mean sea level pressure, vorticity | Single Shot Detector | mAP@IoU=0.5: 79.34-86.64 % |

**Table 2.** Dataset variables.

| Variable | ERA5 attribute name | MERRA-2 attribute name |
|---|---|---|
| sea level pressure | msl | spl |
| precipitable water vapor | tcwv | tqv |
| northward wind at 10 meters | v10 | v10m |
| northward wind at 850 hPa | v850 | v850 |
| eastward wind at 10 meters | u10 | u10m |
| eastward wind at 850 hPa | u850 | u850 |
| temperature at 200 hPa | t200 | t200 |
| temperature at 500 hPa | t500 | t500 |

**Table 3.** The layers of our CNN. The convolutional layer parameter are denoted as <filter size>−<number of filters>, the stride is (1, 1) and no padding was added. The max pooling layer parameters are denoted as <pooling frame>. The fully connected layer parameters are denoted as <number of neurons>. For the activation function of the neurons, "relu" stands for the rectified linear unit whereas "sigmoid" stands for the logistic sigmoid function. Output tensor shapes are also provided for each layer of the CNN for input images of size (16,16,8) and (32,32,8). The number of trainable parameters are 5,053 for images of $16\times16$ px and 30,653 for images of $32\times32$ px.

| Layer type | Parameters | Activation | Output tensor shape for image of $16\times16$ px | Output tensor shape for image of $32\times32$ px |
|---|---|---|---|---|
| convolutional | $3\times3-8$ | relu | 14, 14, 8 | 30, 30, 8 |
| max pooling | $2\times2$ | - | 7, 7, 8 | 15, 15, 8 |
| convolutional | $3\times3-16$ | relu | 5, 5, 16 | 13, 13, 16 |
| max pooling | $2\times2$ | - | 2, 2, 16 | 6, 6, 16 |
| flattening | - | - | 64 | 576 |
| dense | 50 | relu | 50 | 50 |
| dense | 1 | sigmoid | 1 | 1 |

**Table 4.** The layers of the CNN by Liu et al. for comparison with ours. This Table follows the same syntax as Table 3. The number of trainable parameters are 5,776 for images of 16×16 px and 24,976 for images of 32×32 px.

| Layer type | Parameters | Activation | Output tensor shape for image of 16×16 px | Output tensor shape for image of 32×32 px |
|---|---|---|---|---|
| convolutional | 5×5−8 | relu | 12, 12, 8 | 28, 28, 8 |
| max pooling | 2×2 | - | 6, 6, 8 | 14, 14, 8 |
| convolutional | 5×5−16 | relu | 2, 2, 16 | 10, 10, 16 |
| max pooling | 2×2 | - | 1, 1, 16 | 5, 5, 16 |
| flattening | - | - | 16 | 400 |
| dense | 50 | relu | 50 | 50 |
| dense | 2 | sigmoid | 2 | 2 |

**Table 5.** Properties of the image sets. MERRA-2 native is used to construct the other two MERRA-2 image sets, but is not used as input to the CNN.

| Image set | Size (in pixel) | Resolution (in °) |
|---|---|---|
| ERA5 native | 32×32 | 0.25×0.25 |
| ERA5 16px@0.5 | 16×16 | 0.5×0.5 |
| MERRA-2 native | 16×16 | 0.5×0.6 |
| MERRA-2 16px@0.5 | 16×16 | 0.5×0.5 |
| MERRA-2 32px@0.25 | 32×32 | 0.25×0.25 |

**Table 6.** Optimal hyperparameter values.

| Hyperparameter | Value | Search space |
|---|---|---|
| Loss function | Binary cross-entropy | - |
| Training metric | Loss computed on test set | - |
| Maximum number of epochs | 100 | - |
| Early stopping number of epochs ($N$) | 10 | - |
| Batch size | 256 | From 32 to 256, step of 32 |
| Optimizer | Adam | Adam; SGD |
| Learning rate | 0.0001 | 0.0001; 0.001; 0.01 |

**Table A1.** HURDAT2 cyclone categories/status

| Two-letter code | Storm status and Meaning |
|---|---|
| HU | Tropical cyclone of hurricane intensity (> 64 knots) |
| TS | Tropical cyclone of tropical storm intensity (34-63 knots) |
| TD | Tropical cyclone of tropical depression intensity (< 34 knots) |
| EX | Extratropical cyclone (of any intensity) |
| SD | Subtropical cyclone of subtropical depression intensity (< 34 knots) |
| SS | Subtropical cyclone of subtropical storm intensity (> 34 knots) |
| LO | A low that is neither a tropical cyclone, a subtropical cyclone, nor an extratropical cyclone (of any intensity) |
| WV | Tropical Wave (of any intensity) |
| DB | Disturbance (of any intensity) |

**Table A2.** The estimation of the values of the metrics based on iterative cross-validation. The train and test datasets come from the same image set (inter-comparison). The column "Shapiro $p$ on means" refers to the $p$-value of the Shapiro-Wilk test computed on the mean of each iteration, whereas "Shapiro $p$ on all" refers to the $p$-value computed on all the values of the metric.

| Metric | Train dataset | Test dataset | Estimated mean | Estimated std | Shapiro $p$ on means | Shapiro $p$ on all |
|---|---|---|---|---|---|---|
| Accuracy | ERA5 32px@0.25 | same | 0.989748 | 0.002292 | 0.905230 | 1.363663e-04 |
| | ERA5 16px@0.5 | same | 0.989547 | 0.002255 | 0.991043 | 9.576093e-08 |
| | MERRA-2 32px@0.25 | same | 0.982276 | 0.002836 | 0.269320 | 2.560784e-04 |
| | MERRA-2 16px@0.5 | same | 0.981858 | 0.002927 | 0.902361 | 3.120732e-06 |
| AUC | ERA5 32px@0.25 | same | 0.998989 | 0.000643 | 0.088747 | 8.914557e-12 |
| | ERA5 16px@0.5 | same | 0.998936 | 0.000638 | 0.506771 | 2.956014e-11 |
| | MERRA-2 32px@0.25 | same | 0.997114 | 0.001140 | 0.017086 | 3.536823e-14 |
| | MERRA-2 16px@0.5 | same | 0.996904 | 0.001107 | 0.239811 | 7.852716e-14 |
| AUPRC | ERA5 32px@0.25 | same | 0.998430 | 0.000811 | 0.046393 | 2.159975e-09 |
| | ERA5 16px@0.5 | same | 0.998374 | 0.000796 | 0.359663 | 4.650617e-11 |
| | MERRA-2 32px@0.25 | same | 0.995573 | 0.001183 | 0.274620 | 3.556242e-12 |
| | MERRA-2 16px@0.5 | same | 0.995337 | 0.001319 | 0.769673 | 3.934050e-13 |

**Table A3.** Comparisons of metric values of models taken two by two (inter-comparisons). The column "Kruskal $p$-value" refers to the $p$-value of the Kruskal-Wallis test computed on all the values of the metrics. The column "Comparable" indicates whether the null hypothesis is accepted for an alpha level of 1 %.

| Metric | Train/test dataset | Train/test dataset | Kruskal $p$-value | Comparable |
|--------|--------------------|--------------------|-------------------|------------|
| Accuracy | ERA5 32px@0.25/same | ERA5 16px@0.5/same | 9.173333e-01 | True |
| | ERA5 32px@0.25/same | MERRA-2 32px@0.25/same | 8.451539e-31 | False |
| | ERA5 32px@0.25/same | MERRA-2 16px@0.5/same | 2.367641e-33 | False |
| | ERA5 16px@0.5/same | MERRA-2 32px@0.25/same | 9.721777e-29 | False |
| | ERA5 16px@0.5/same | MERRA-2 16px@0.5/same | 9.251784e-31 | False |
| | MERRA-2 32px@0.25/same | MERRA-2 16px@0.5/same | 5.506329e-01 | True |
| AUC | ERA5 32px@0.25/same | ERA5 16px@0.5/same | 5.243858e-01 | True |
| | ERA5 32px@0.25/same | MERRA-2 32px@0.25/same | 5.901992e-26 | False |
| | ERA5 32px@0.25/same | MERRA-2 16px@0.5/same | 9.964538e-32 | False |
| | ERA5 16px@0.5/same | MERRA-2 32px@0.25/same | 5.982768e-24 | False |
| | ERA5 16px@0.5/same | MERRA-2 16px@0.5/same | 6.935282e-30 | False |
| | MERRA-2 32px@0.25/same | MERRA-2 16px@0.5/same | 1.174539e-01 | True |
| AUPRC | ERA5 32px@0.25/same | ERA5 16px@0.5/same | 7.391314e-01 | True |
| | ERA5 32px@0.25/same | MERRA-2 32px@0.25/same | 2.082609e-30 | False |
| | ERA5 32px@0.25/same | MERRA-2 16px@0.5/same | 2.401478e-35 | False |
| | ERA5 16px@0.5/same | MERRA-2 32px@0.25/same | 1.952581e-29 | False |
| | ERA5 16px@0.5/same | MERRA-2 16px@0.5/same | 3.139588e-34 | False |
| | MERRA-2 32px@0.25/same | MERRA-2 16px@0.5/same | 2.450381e-01 | True |

**Table A4.** The estimation of the values of the metrics based on iterative cross-validation. The train and test datasets come from different image set, but have the same image resolution and size (cross-comparisons). The column "Shapiro $p$ on means" refers to the $p$-value of the Shapiro-Wilk test computed on the mean of each iteration, whereas "Shapiro $p$ on all" refers to the $p$-value computed on all the values of the metric.

| Metric | Train dataset | Test dataset | Estimated mean | Estimated std | Shapiro $p$ on means | Shapiro $p$ on all |
|---|---|---|---|---|---|---|
| Accuracy | ERA5 32px@0.25 | MERRA-2 32px@0.25 | 0.977677 | 0.003182 | 0.226578 | 0.017054 |
| | ERA5 16px@0.5 | MERRA-2 16px@0.5 | 0.977721 | 0.002321 | 0.786720 | 0.013096 |
| | MERRA-2 32px@0.25 | ERA5 32px@0.25 | 0.986523 | 0.002604 | 0.073408 | 0.193609 |
| | MERRA-2 16px@0.5 | ERA5 16px@0.5 | 0.986655 | 0.002943 | 0.540750 | 0.019739 |
| AUC | ERA5 32px@0.25 | MERRA-2 32px@0.25 | 0.995235 | 0.001789 | 0.762134 | 5.846574e-09 |
| | ERA5 16px@0.5 | MERRA-2 16px@0.5 | 0.995413 | 0.001208 | 0.871147 | 2.561682e-12 |
| | MERRA-2 32px@0.25 | ERA5 32px@0.25 | 0.998357 | 0.000803 | 0.719871 | 1.581214e-12 |
| | MERRA-2 16px@0.5 | ERA5 16px@0.5 | 0.998323 | 0.000929 | 0.215594 | 5.414141e-09 |
| AUPRC | ERA5 32px@0.25 | MERRA-2 32px@0.25 | 0.993296 | 0.001898 | 0.436782 | 7.346610e-08 |
| | ERA5 16px@0.5 | MERRA-2 16px@0.5 | 0.993437 | 0.001331 | 0.905712 | 2.634040e-11 |
| | MERRA-2 32px@0.25 | ERA5 32px@0.25 | 0.997537 | 0.000852 | 0.825225 | 2.476906e-09 |
| | MERRA-2 16px@0.5 | ERA5 16px@0.5 | 0.997549 | 0.001092 | 0.694913 | 1.416567e-07 |

**Table A5.** The values of the metrics from models trained and tested on the same image set, compared to those from models trained on one image set and tested on the other (cross-comparison). The column "Kruskal $p$-value" refers to the $p$-value of the Kruskal-Wallis test computed on all the values of the metrics. The column "Comparable" indicates whether the null hypothesis is accepted for an alpha level of 1 %.

| Metric | Train/test dataset | Train/test dataset | Kruskal $p$-value | Comparable |
|---|---|---|---|---|
| Accuracy | ERA5 32px@0.25/same | ERA5 32px@0.25/MERRA-2 32px@0.25 | 4.857791e-47 | False |
| | ERA5 16px@0.5/same | ERA5 16px@0.5/MERRA-2 16px@0.5 | 1.005262e-44 | False |
| | MERRA-2 32px@0.25/same | MERRA-2 32px@0.25/ERA5 32px@0.25 | 1.825678e-11 | False |
| | MERRA-2 16px@0.5/same | MERRA-2 16px@0.5/ERA5 16px@0.5 | 1.434371e-14 | False |
| AUC | ERA5 32px@0.25/same | ERA5 32px@0.25/MERRA-2 32px@0.25 | 4.011721e-46 | False |
| | ERA5 16px@0.5/same | ERA5 16px@0.5/MERRA-2 16px@0.5 | 2.698731e-43 | False |
| | MERRA-2 32px@0.25/same | MERRA-2 32px@0.25/ERA5 32px@0.25 | 7.645103e-09 | False |
| | MERRA-2 16px@0.5/same | MERRA-2 16px@0.5/ERA5 16px@0.5 | 2.074129e-12 | False |
| AUPRC | ERA5 32px@0.25/same | ERA5 32px@0.25/MERRA-2 32px@0.25 | 1.640760e-48 | False |
| | ERA5 16px@0.5/same | ERA5 16px@0.5/MERRA-2 16px@0.5 | 3.619129e-47 | False |
| | MERRA-2 32px@0.25/same | MERRA-2 32px@0.25/ERA5 32px@0.25 | 3.864565e-12 | False |
| | MERRA-2 16px@0.5/same | MERRA-2 16px@0.5/ERA5 16px@0.5 | 1.332830e-16 | False |

**Table A6.** Statistics of failed predictions by combinations of training/testing datasets.

| Training dataset | Test dataset | Specs | Total failed | False negatives | False positives |
|---|---|---|---|---|---|
| ERA5 | ERA5 | 32px@0.25 | 68 (0.88 %) | 44 | 24 |
| ERA5 | MERRA-2 | 32px@0.25 | 156 (2.02 %) | 125 | 31 |
| ERA5 | ERA5 | 16px@0.5 | 73 (0.94 %) | 46 | 27 |
| ERA5 | MERRA-2 | 16px@0.5 | 155 (2.00 %) | 128 | 27 |
| MERRA-2 | MERRA-2 | 32px@0.25 | 110 (1.42 %) | 73 | 37 |
| MERRA-2 | ERA5 | 32px@0.25 | 93 (1.20 %) | 58 | 35 |
| MERRA-2 | MERRA-2 | 16px@0.5 | 100 (1.29 %) | 52 | 48 |
| MERRA-2 | ERA5 | 16px@0.5 | 90 (1.16 %) | 40 | 50 |

**Table A7.** Background images wrongly classified as TC-containing images (false positives) for all combinations of training/testing datasets. For each image we also indicate the status of the image according to HURDAT2 if present in the database ("None" if not present), the probability of the classification (with its standard deviation across the combinations of training/testing datasets), and the temporal distance to a cyclone in the past and in the future (with the status of the cyclone).

| #index | Status | Mean prob | Past (hours) | Future (hours) | HURDAT2 id |
|---|---|---|---|---|---|
| 207 | SD | 0.9905±0.0205 | 8250 (HU) | 90 (TS) | AL061990 |
| 3149 | None | 0.8819±0.0544 | 1308 (TS) | 354 (TS) | AL122008 |
| 4163 | WV | 0.9919±0.0150 | 174 (TS) | 228 (TS) | AL092012 |
| 6048 | None | 0.8240±0.1111 | 132 (TS) | 54 (TS) | AL071998 |
| 6059 | EX | 0.9963±0.0047 | 228 (TS) | 96 (TS) | AL132018 |
| 6295 | None | 0.8918±0.1072 | 162 (HU) | 60 (HU) | AL132003 |
| 6836 | None | 0.9216±0.0542 | 168 (TS) | 90 (TS) | AL162000 |

**Table A8.** The single TC-contaning image wrongly classified as background (false negative) for all combinations of training/testing datasets. The status of the image according to HURDAT2, the probability of the classification (with its standard deviation across the combinations of training/testing datasets) are indicated. The columns past and future reflect the cyclonic activity in the geographical area of the image, i.e. the temporal distance to the first and the last tracks of the cyclone (and their HURDAT2 status).

| #index | Status | Mean prob | Past (hours) | Future (hours) | HURDAT2 id |
|---|---|---|---|---|---|
| 290 | TS | 0.0855±0.0665 | 72 (bckgrd) | 6 (TD) | AL162000 |