# Peer review of "Classification of tropical cyclone containing images using a convolutional neural network: performance and sensitivity to the learning dataset"

_EGUsphere, 2022_

## Author Response (AR1)

Dear Reviewers,

First of all, we would like to thank you for your contribution to the improvement of our article. We make below a point-by-point answer to your remarks.

We wish you a good receipt of our modified manuscript.

Regards,

Sébastien Gardoll and Olivier Boucher

**Reviewer 1**

**Section 1**

- This study uses a very simple CNN design with two convolutional layers based on a paper from 2016. CNN research has progressed dramatically since then and more sophisticated architectures with much better performance for image classification now exist. Many of these have already found their way into atmospheric studies. While there is nothing wrong with using a simple model (and the simple model performs quite well in this case) the last 8-years of CNN research should probably be acknowledged in the background section.

We agree with the Reviewer that the field has moved since 2016. We have performed a literature review (see Table below) for TC detection and segmentation in both satellite and model data, which we have also included in the Introduction section of the revised manuscript. The Table shows that a range of architectures including ad hoc CNN, YOLO, U-Net, DeepLabv3+ have been used over the last 8 years. The bibliography of the revised manuscript has also been completed. The Reviewer is also right in saying that the simple models perform quite well, hence it is not clear that better performance can be obtained with the most complex architectures, and the simple models remain relevant for our study that focuses on cross-training of the CNN on different sets of reanalyses and the use of a wider set of performance metrics than in previous studies.

We have added the following lines in the Section 1:

"There is also a wealth of studies on the detection of TCs in satellite imagery, reanalysis and climate model outputs based on machine learning (ML) approaches. Table 1 summarizes notable studies published in the last eight years that implement neural architectures based on convolution layers. It is not surprising that this approach was favored because TCs have very distinct features which make them relatively easy to detect with convolutional neural networks. Since Liu et al. (2016), whose deep learning (DL) model only classifies pre-existent images, various subsequent studies have focused on improving the detection of all cyclones at once present in unidimensional or multidimensional meteorological images (e.g., Ebert-Uphoff and Hilburn, 2020) and climate model data (e.g., Matsuoka et al., 2018).

This latter work focuses on the detection of cyclones using a CNN image classifier which operates on a sliding window of output from the Nonhydrostatic Icosahedral Atmospheric Model (NICAM) and studies the system performance in terms of detectability. The detection can be either "coarse" by drawing rectangular envelopes around the cyclones (the studies are flagged as detection in the Purpose column of the Table) or "precise" by drawing the contours of the cyclones including their internal structure (studies flagged as segmentation). The main idea of these more recent studies is to apply new DL model architectures coming from computer vision research (e.g., U-Net, DeepLabv3, YOLOv3, Single Shot Detector, etc.) to the analysis of meteorological features such as cyclones."

| Authors | Year | Purpose | Dataset | Variables | NN Architecture | Performance at best |
|---|---|---|---|---|---|---|
| Liu *et al.* | 2016 | Cyclone image classification | ERA-interim (rea) CAM5.1 (mod) NCEP-NCAR (rea) 20th Century Reanalysis | Pressure sea level, wind vectors at 10 meters and at 850 hPa, temperature at 200 and 500 hPa, total column water vapour | ad hoc CNN | Accuracy: 99 % |
| Racah *et al.* | 2016 | Cyclone segmentation | CAM5 (mod: 25 km) | The 16 channels of CAM5 | ad hoc autoencoder | mAP@IoU=0.1: 52.92 % |
| Hong *et al.* | 2017 | Eye detection | COMS-1 (sat) | 4 IR channels | GoogLeNet | RMSE: 0.02 |
| Matsuoka *et al.* | 2018 | Detection of cyclone by sliding window and cyclone image classifier | NICAM (mod: 14 km) + NSW6 + HadISST | Outgoing longwave radiation | ad hoc CNN | Probability of detection: 79.9–89.1 % false alarm ratio: 32.8–53.4 % |
| Kumler-Bonfanti *et al.* | 2020 | Cyclone segmentation | GFS (mod) | Total precipitable water | U-Net | Accuracy: 0.991 Dice coeff: 0.763 Tversky coeff: 0.75 |
| Shakya *et al.* | 2020 | Cyclone image classification | KALPANA-I (sat) MOSDAC (sat) | IR, Vis | ad hoc CNN | Accuracy: 97 % |
| Shakya *et al.* | 2020 | Cyclone detection and path prediction | KALPANA-I (sat) MOSDAC (sat) | IR, Vis | RetinaNet and polynomial regression | RMSE: 5-15.55 % |
| Prabhat *et al.* | 2020 | Cyclone segmentation | CAM5.1 (mod: 25 km) | Integrated vapor transport, integrated water vapor, vorticity, | DeepLabv3+ | IoU: 0.2441 |

| | | | | wind vectors at 10 meters and at 850 hPa, sea level pressure | | |
|---|---|---|---|---|---|---|
| Pang *et al.* | 2021 | Cyclone detection | Satellite images from NII | Vis | DCGAN and YOLOv3 | Accuracy: 97.78 % mAP@IoU=0.5: 81.39 % |
| Shi *et al.* | 2022 | Extratropical cyclone detection | ERA5 (rea) | Top net thermal radiation, mean sea level pressure, vorticity | Single Shot Detector | mAP@IoU=0.5: 79.34-86.64 % |

Table. Summary of previous studies aiming at the detection or the segmentation of TC in satellite or model data. coeff: coefficient; IoU: intersection over union; IR: infrared; mAP: mean average precision; mod: model; rea: reanalysis; RMSE: root mean square error; sat: satellite; vis: visible.

**Section 2**

- Lack of large training data and potential difficulties applying ML models to new datasets are both concerns/points mentioned in this study. Data augmentation schemes and model regularization techniques like dropout and batch normalization are proven ways to improve the robustness of CNN models when applied to new datasets and in the face of limited training data. These have become so ubiquitous that almost all CNN-based image recognition studies now use at least one of them. I think not using or at the very least acknowledging the existence of these methods is a major omission, since they are both very common in the literature and very relevant to this specific problem.

We agree with the Reviewer that data augmentation schemes and model regularization techniques are important. They are not necessarily needed in this study because the performance of our retrievals is already very good. However they could become important in the next phase when we seek to detect TC in climate model simulations with a CNN trained on a reanalysis dataset. We have now mentioned these techniques in the discussion Section and justify why we do not require them at this stage:

"Data augmentation (especially geometric transformations; Shorten and Khoshgoftaar, 2019) and model regularization techniques (e.g., weight-decay, batch normalization, dropout, etc.) are proven ways to improve the robustness of a CNN trained with a dataset of limited size. Our dataset contains 26,954 images, which is relatively small compared to the size of datasets encountered in many computer vision applications (for instance, Imagenet contains more than 14 million images). However using these techniques was not justified for our study because the performance of our CNN without data augmentation is already very high. Such techniques could however become very relevant in future work when we seek to detect TC in climate model simulations with a CNN trained on a reanalysis dataset. Indeed different climate models may simulate TC imperfectly and there is probably some value in offering a larger variety of TC structures to the training dataset. It is expected that the simulation of TCs increases in quality with the climate model resolution (Strachan et al., 2013) and climate models running at resolutions of 10 to 50 km are now commonplace. Likewise we would need to augment the number of images with very intense TC or TC migrating outside their usual domains because there are indications that such situations may become more frequent with global warming and we want to ensure these can be detected adequately in climate simulations."

New references added:
Shorten, C., Khoshgoftaar, T.M. A survey on image data augmentation for Deep Learning. *J Big Data*, 6, 60, 2019. doi:10.1186/s40537-019-0197-0

Strachan, J., P.L. Vidale, K. Hodges, M. Roberts, and M. Demory. Investigating global tropical cyclone activity with a hierarchy of AGCMs: The role of model resolution, *Journal of Climate*, *26*(1), 133-152, 2013. doi: 10.1175/JCLI-D-12-00012.1

- The use of the term "images" to refer to reanalysis data may be a bit confusing, particularly for those not familiar with reanalysis but perhaps familiar with CNNs which are often applied to photos. I think it is fine to use but it should be addressed early in the manuscript that "images" here does not refer to photographs, optical data, or data from any type of imager for that matter. Instead, these are chunks of regularly gridded data extracted from a numerical simulation.

We agree with the Reviewer, we have added the following text in Subsection 2.3:

"2.3 Images

In computer vision, the term image refers to a stack of matrices (also called a 3D tensor), with each matrix representing an information channel. For example, RGB images are formed of a stack of matrices of numerical values coding the red (R), green (G) and blue (B) color intensities of each pixel of a photograph. Our use of the term image is a generalization of the concept of RGB images. In the rest of our study, an image refers to a stack of gridded data extracted from a different variable of ERA5 or MERRA-2 on a given geographical area. Unlike for an RGB image, the channels cannot be combined; we thus graphically represent each channel separately."

- There is some discussion of ensuring that input data has been interpolated to the correct grid resolution for consistent inputs to the CNN, but is there any concern about the non-uniform grid spacing on a lat lon grid? Samples closer to the poles will cover a smaller area.

We believe this is not a major problem as we are concerned with cyclones in the tropical region where the deformation induced by a regular lat-lon grid is small (the area of a surface element in spherical coordinates is proportional to cos(lat)). The Reviewer has nevertheless a very valid point for the detection of TC that migrate polewards and mid-latitude depressions as the grid spacing would matter more at mid-latitudes. This is also an interesting point because data augmentation methods may help to account for such deformation as a function of latitude and therefore could be used to increase the robustness of the CNN. We now discuss this in the Discussion section, just after the previous addition about data augmentation:

"In this study we work on images created on a regular lat-lon grid, which potentially introduces a deformation because of the cos(latitude) dependence of a displacement element along the longitude. Such a deformation is small in the tropical region and therefore is not thought to be a problem for our analysis. However it increases as a function of latitude, so it may become an important factor to consider for TC that migrate polewards or for the detection of mid-latitude depressions. Data augmentation techniques that introduce deformed images in the training datasets could help to increase the robustness of the CNN in these situations."

- Line 85: Only 5 variables are listed. You should mention that you are using winds and temps at two levels which gets you to 8 input variables.

We agree, we have modified the following sentence: "We use fields of sea level pressure, precipitable water vapor, the two components of the wind (at the surface and at 850 hPa) and the temperature at two different pressure levels (see Table 2)."

- Line 85: I think these input variables are very reasonable choices, but no motivation was given for why these specific ones were used.

Indeed, these input variables represent a very reasonable choice. First of all, we based our study on the same variables used by Liu *et al.* (2016). The choice of most of these variables is confirmed by Racah *et al.* (2017), Prabhat *et al.* (2020), as well as Kumler-Bonfanti *et al.* (2020) concerning the precipitable water and Shi *et al.* (2022) concerning the mean sea level pressure. This latter study also considers the vorticity, which is a function of the wind field (see the neuronal network architectures in the Table above). It is likely that there is redundant information in all these variables. We recognize that quantifying the relative contributions of these variables in the classification decision of the CNN would be interesting, with the aim of reducing the number of variables. It is in this perspective that we have written our potential future work section.

We also added the following lines in Section 2.2 "Meteorological reanalyses":

"We have followed Liu et al. (2016) and considered an extensive set of meteorological variables to detect TC (see Table 2). This choice is confirmed by subsequent studies (Racah *et al.*, 2017; Prabhat *et al.*, 2020; Kumler-Bonfanti *et al.*, 2020). It is likely that there is redundant information in this set of variables. An interesting follow-up work will be to investigate the relative contributions of these variables in the classification decision of the CNN, with the aim of reducing the number of variables."

**Section 3**

- Section 3.2.3: It seems that a great deal of effort has been put into identifying non-cyclone training samples that thoroughly cover the region where cyclones have been observed. However, this appears to exclude any negative samples from other parts of the North Atlantic basin. Including negative samples from the Eastern Atlantic or the extratropical storm track region is important if the algorithm will ultimately be applied to the entire basin to ensure there is not unexpected behavior in these areas. The background section mentioned the potential of TCs moving farther north in the Atlantic in a warming climate.

This is a valid point but out of scope for this study. It is relatively easy to add non-TC images further north in the basin. It is more complicated to add TC images further north as there are fewer TC on the edge of the basin. This is another area where data augmentation methods could help. We now discuss this in the Discussion section (text repeated from our reply above):

"Likewise we would need to augment the number of images with very intense TC or TC migrating outside their usual domains because there are indications that such situations may

become more frequent with global warming and we want to ensure these can be detected adequately in climate simulations."

- Section 3.3 and 3.4: I was left confused about exactly how the training was performed, and I think more details should be included to improve reproducibility. Here are some notes:
    - Information like the loss function and optimizer used are more important to include in the main text than details about the hardware used. This information is key to reproducibility, but the hardware is not.

We agree with the Reviewer. We now present this information as a Table next in the main text. Table A3 has been moved to Section 3.3.

- Is the early stopping based on training loss (line 203) or test set loss (table A3 line 2)? Is there evidence of overfitting without early stopping?

We observe a slight overfitting with the growing number of training epochs, which is why we use the early stopping method that corresponds more or less to the elbow method. The early stopping method is based on the test set loss, which is not biased. We have added the information in Section 3.3 on model training:

"Overfitting has been noticed during the training of the model. We have observed the characteristic U-shape of underfitting followed by overfitting by plotting the value of the lost function calculated using the validation dataset against the number of epochs. In order to automatically avoid overfitting, we used two Tensorflow callbacks: early stopping and model check point. The first callback stops the training after N epochs without further improving the training metric (N is set to a value of 10). Early stopping behaves more or less like the elbow method."

- Which dataset was used for the hyperparameter optimization (HPO)?
- The exact range of parameters tested during HPO is not provided
- What kind of train/test/validation split was used for the HPO process? Was a separate test set held out during hyperparameter tuning to avoid an overfit?

First of all, we would like to clarify our intention regarding the optimization of the hyperparameters: while we follow the approach by Liu *et al.* (2016), these authors do not provide the values of the training hyperparameters such as batch size, optimizer and learning rate. Instead of fixing these values in an arbitrary way, we searched for local optimal values of these hyperparameters to maximize the performance of the CNN. However, these values depend on the image dataset being used (ERA5 32px@0.25, MERRA-2 32px@0.25, etc.) and the way it is split between the classical training/validation/test sub-datasets. The training and validation datasets are used for the optimization phase while the test dataset is used for the unbiased evaluation of the performance. We conducted four optimizations for the different image datasets but for the same split (0.7/0.15/0.15) and we obtained the same values for the optimizer and the learning rate with very similar performances. Only the batch size differs among these four optimizations, so we decided to set a value as large as possible given the memory of the GPU cards at our disposal. Of course, these optimal

values are only valid for the given split, however we think that they are close to the global optimum, because the performances vary very little according to the different values of these hyperparameters. But above all, by fixing the same hyperparameter values for all our numerical experiments, we avoid attributing the variability of the studied metrics to hyperparameter changes. The range of hyperparameters is added in a new column called search space. The ranges are based on values commonly found in the literature.

We have added these aspects in the revised manuscript:

"Our work is based on the study by Liu *et al*., but these authors did not provide the values of their training hyperparameters such as batch size, optimizer and learning rate. Instead of fixing these values in an arbitrary way, we search for local optimal values of these hyperparameters to maximize the performance of the CNN. Since training times are relatively short on our GPU cluster, we performed a grid search hyperparameter optimization to maximize the score of the training metric, using conventional hyperparameter value ranges (the number of combinations of the search space is 48). We conducted four optimizations for the different image datasets but for the same training / validation / testing split (0.70/0.15/0.15). We obtained the same values for the optimizer and the learning rate, for very close performances. Only the batch size differs, so we decided to set a value as large as possible given the memory of the GPU cards at our disposal. Of course, these optimal values are only valid for the given split, however we think that they are close to the global optimum, because the performances vary very little according to the different values of these hyperparameters."

- Section 3.4: This section would benefit from a re-write. Currently, it introduces how k-fold cross validation is typically performed only to then say you used a different procedure, then adds that you actually train 20 models with 1/10th sized test sets. Perhaps open by describing precisely what you did and follow with the motivations for some of your decisions (dealing with temporal autocorrelation and increasing the number of samples). Also, k-folds is a common ML technique. You could just cite a reference here to avoid some of the clutter of talking about typical k-folds strategies that weren't used in this paper, e.g. Bishop, C. M., 2006, "Pattern Recognition and Machine Learning", Ch. 1.

In response to the Reviewer's remark, we have rewritten the description of the evaluation of the metrics as following:

"For the evaluation and comparison of the metrics (developed in Section 4.2), we wanted to be able to calculate the expected value and the uncertainty of the metrics, without bias. To that end, we applied an iterative cross validation method which consists in repeating 20 times a cross validation method. We chose the k-fold method (Bishop, 2006), with k equal to ten, as the cross validation method. We obtained a mean of the metrics for each k-fold iteration. By applying the central limit theorem on this set of metric means, we could compute the expected value and the uncertainty of the metrics.

In order to avoid any bias, we took care to check if the central limit theorem can be applied, by testing the normality of the distribution of the metric means using the Shapiro-Wilk statistical test (brief non-mathematical presentation given in Appendix B1). Moreover,

images coming from a time series of tracks from the same cyclone may be found in both the training and test datasets, which would induce some dependance between the training and test datasets due to the autocorrelation within individual cyclone tracks. In order to avoid such a bias, the k-fold split is based on sampling the years randomly and balancing the folds as much as possible. The partitioning combinations are calculated in advance in order to guarantee the uniqueness of their composition. Scale bias is also avoided by standardizing the channels of the images online, just before training the CNN.

Finally, for the comparison of the metric means, we chose to apply the Kruskal-Wallis statistical test (brief non-mathematical presentation given in Appendix B2) for an alpha level of 1 %, because the Shapiro-Wilk test was negative for most distributions of metric values of our experiments, invalidating the use of the Student's t-test.

For the experiment of highlighting the problem with the accuracy (point developed in Section 4.1), we applied the classical hold-out method, avoiding the autocorrelation between images belonging to a same cyclone track, with the following partitioning: 70 % of the data for the training dataset and 30 % of the data for the test dataset."

- Line 241: It would be helpful to move this note about standardizing the input variables into the "data" section. Or Section 3.2 "image preparation"

We agree to mention image preparation, the standardization of the data of each channel taken separately in Section 3.2, as it is a necessity for the training of the CNN.

The following subsection has been added:

"3.2.7 Data standardization

Neural network models learn a mapping from input variables to an output variable. The input variables have nearly always different scales and large scale differences are detrimental to the learning process of neural networks. In order to ensure that each variable is equally important, whatever its range of values, input variables are rescaled to the same scale. There are several methods such as standardization (or Z-score normalization) which consists in recalculating the values of the variables so that their mean and standard deviation equal to zero and one, respectively. In our study, we have systematically standardized each channel of the images, by calculating the means and standard deviations of the channels on all the images of the training set. The validation and test image datasets are excluded from the calculation of the mean and standard deviation, to avoid that information about the validation and test datasets leak into the training phase. However the validation and test datasets are also scaled using the mean and standard deviation of the training dataset."

Moreover it is also necessary to specify the modality of this standardization in the context of the iterative k-fold method. We therefore felt that line 241 belongs better in Section 3.4 "Evaluation of metrics", just after the presentation of the iterative k-fold method.

- The authors justify use of the 2-convolution layer model by saying it will help prevent overfitting, but no analysis is done to back up this argument. Is overfitting actually a problem?

Indeed, overfitting is observed during the training of the CNN for a significant number of epochs. When plotting the value of the lost function (calculated on the validation dataset) against the number of epochs (using the software Tensorboard), we observe the characteristic U-shape of underfitting followed by overfitting. The overfitting was automatically prevented using the early stopping callback of the Keras programming library that acts as the elbow method.

**Section 4**

- Maybe I missed it, but I don't think the "simple models" shown in Figure 7 are described anywhere. Also, the legend calls them "metrics" is it supposed to say "models"?

The Reviewer is quite right. The legend of Figure 7 has been fixed and we have added the following explanation about the simple models in Section 4.2.1:

"Indeed, the usefulness of a model is measured by the difference between its performance and that of models based on simple rules or a domain specific baseline. For instance, we implement the following simple models (from the software library scikit-learn): "most frequent" which always predict the most frequent class observed in the training dataset (i.e., background), "stratified" which generates randomly predictions at probabilities that respect the class distribution of the training dataset (i.e., 1/3 cyclone, 2/3 background), "uniform" which generates predictions uniformly at random background or cyclone with equal probability."

We removed the "constant" and "prior" models that behave identically to "most frequent" in our experiment.

**Section 5**

- I remain skeptical of the claim in the conclusions section that the lower performance of the CNN on MERRA-2 implies that MERRA has lower information content for TC identification. I suppose one can argue that ERA5 has more information content simply because it is higher resolution, but that additional information may not improve cyclone detection. Only one CNN was tested, and perhaps a larger CNN, different input fields, different training procedure, data augmentation, etc. would yield better results on the MERRA data. Are all the storms present in HURDAT2 present in MERRA? If so, one could likely design a model better suited for the MERRA dataset. I think a more reasonable claim is made at the end of Section 4.2.2: "Thus we can conclude that the ERA5 dataset is more information rich than the MERRA-2 dataset for the classification of cyclone images *using our CNN.*"

We agree with the Reviewer that our conclusion about MERRA-2 having less information content for TC image classification than ERA5 should be qualified and we adopt the rephrasing proposed by the Reviewer. The following lines have been added to the conclusion:

"Applying an ERA5-trained CNN on MERRA-2 images works better than applying a MERRA-2 trained CNN on ERA5 images, which suggests that ERA5 has a larger information content in the framework of our CNN.

This is also consistent with the findings of Malakar *et al.* (2020) who analyzed the error in the location of the center, maximum winds and minimum pressure at sea level in six meteorological reanalyses including ERA5 and MERRA-2 for the evolution of 28 TCs occurring between 2006 and 2015 over the North Indian Ocean, with respect to the observations of the Indian Meteorological Department (IMD). The authors of this study show, among other things, that the ERA5 dataset captures the evolution of these TCs in a more realistic way than MERRA-2 (i.e., smaller errors in the previous variables). They also show that ERA5 and MERRA-2 can capture the intensity of the TCs from the depression stage to the very severe cyclonic storm stage but not from the extremely severe cyclonic storm stage for which the intensity of the TCs is underestimated. However, they conclude that of the six datasets, ERA5 provides the best representation of the TC structure in terms of intensity. Finally, the study published by Hodges *et al.* (2017) shows that 95 % of the Northern Hemisphere TC tracks, from the IBTrACS database that includes HURDAT2, are present in MERRA-2. Unfortunately, this study does not include ERA5. It also confirms the underestimation of cyclone intensity in MERRA-2 compared to observations."

New references added:

Hodges, K., Cobb, A., & Vidale, P. L. (2017). How well are tropical cyclones represented in reanalysis datasets?, *Journal of Climate*, *30*(14), 5243-5264.

Malakar, P., Kesarkar, A. P., Bhate, J. N., Singh, V., & Deshamukhya, A. (2020). Comparison of reanalysis data sets to comprehend the evolution of tropical cyclones over North Indian Ocean. *Earth and Space Science*, 7, e2019EA000978. doi: 10.1029/2019EA000978

**Reviewer 2**

**Section 1**

- In abstract, clarify that this work is based on TCs in the North Atlantic only.
- Line 10: this sentence sounds incorrect
- The motivation of this study is not clear. I think the main purpose of this study is not simply showcasing the performance of CNN in TC detection—but examining the sensitivity of CNN-based TC detection algorithm to input and training datasets (as written in L49-50 and L53-54) and/or presenting a new way to preparing the training datasets for TC detections (as shown in figures 2 and 3). However, this point is not effectively highlighted in the introduction; in fact, it is not mentioned at all. My suggestion is to restructure the introduction to make clear the purpose of this study and also state the main purpose of this study clearly in the abstract.

We have restructured and strengthened the introduction to better explain the motivation of the study.

We also restructured the abstract:

"Tropical cyclones (TCs) are one of the most devastating natural disasters [...] This study compares the performance and sensitivity of a CNN to the learning dataset. For this purpose, we chose two meteorological reanalysis, ERA5 and MERRA-2, and used a number of meteorological variables from them to form TC-containing and background images..."

- (L27-29; L347-389) The authors claim this work contributes towards "automatic detection of TC in climate simulations without the need to retrain the CNN for each new climate model or climate model resolution." However, this point needs to be further justified. First, the cross examination (e., training on dataset A and evaluating on dataset B) would give more consistent results for two same-generation reanalyses than for two different climate models. Reanalysis datasets are at least grounded by observations; but two different climate models can wildly differ (*e.g.*, dynamical core, subgrid schemes, and coupling between components). Second, the current study examined only 0.25° and 0.5° resolutions which is much finer than the majority of CMIP-class climate models (~1°). The authors test interpolation from a finer to a coarse resolution, and accordingly, some high-resolution information is still carried over to interpolated low-resolution fields. However, the native resolution of common climate models is already too coarse (~1°) and their output variables calculated on a coarse grid are likely to miss finer dynamics/physics. In this regard, the author's CNN algorithm may not work well for climate model applications.

The Reviewer is right that we do not demonstrate that we can train the CNN on reanalysis and successfully retrieve TC in climate model simulations. This is certainly fraught with difficulties but this remains our objective in the near future.

We agree that TC are not simulated or at least not well simulated at resolutions less than 100 km (see Strachan *et al.*, 2013). However climate models running at resolution of 10 to 50 km are now commonplace (see e.g. HighResMIP and references in our manuscript). We do not argue that the CNN algorithm should be applied to 1° resolution climate models but only to the high-resolution models. Data augmentation methods (as discussed in the response to Reviewer #1's comments) may also help in this respect. We have strengthened the discussion section of the revised manuscript in that respect. The additional text reads:

"... Such [augmentation] techniques could however become very relevant in future work when we seek to detect TC in climate model simulations with a CNN trained on a reanalysis dataset. Indeed different climate models may simulate TC imperfectly and there is probably some value in offering a larger variety of TC structures to the training dataset. It is expected that the simulation of TCs increases in quality with the climate model resolution (Strachan et al., 2013) and climate models running at resolutions of 10 to 50 km are now commonplace. Likewise we would need to augment the number of images with very intense TC or TC migrating outside their usual domains because there are indications that such situations may become more frequent with global warming and we want to ensure these can be detected adequately in climate simulations."

Furthermore we have started to collaborate with colleagues (S. Bourdin and S. Fromang) in our institute who have applied a physical detection of TC in a high-resolution version of the IPSL climate model (see Bourdin *et al.*, 2022).

Additional references:

Bourdin, S., S. Fromang, W. Dulac, J. Cattiaux, and F. Chauvin, Intercomparison of four tropical cyclones detection algorithms on ERA5, *EGUsphere* [preprint], https://doi.org/10.5194/egusphere-2022-179, 2022.

Strachan, J., P.L. Vidale, K. Hodges, M. Roberts, and M. Demory. Investigating global tropical cyclone activity with a hierarchy of AGCMs: The role of model resolution, *Journal of Climate*, *26*(1), 133-152, 2013.  doi: 10.1175/JCLI-D-12-00012.1

- (L30-40) The physical algorithms require preset thresholds. It provides intrinsic weakness because the performance of a given algorithm would depend on those thresholds. However, CNN classification (and, generally, any ML approaches) also suffer from similar issues. The performance of CNN is sensitive to the choice of hyperparameters. Unlike the thresholds used in physical algorithms (which can be physically interpreted), we do not know how each of these hyperparameters affects the performance of neural networks. As the thresholds in a physical algorithm is tuned, the hyperparameters of neural networks are tuned. In this view, the ML approach does not solve the problems related to thresholds in a physical mechanism. Besides, the authors point out that physical algorithms are *usually* applied in a limited domain. I wonder if it is due to the incapability of such algorithms to be applied in a wider domain—or if it is simply that people have not tried to apply it for a wider domain. If the latter is the case, the ML approach does not really address the issue of limited domains.

The Reviewer is right that ML approaches do not bypass the need to "tune" the retrieval models. Like for physical algorithms, some of this tuning can however be done in a more or less objective way (see our response to Reviewer #1).

Both physical and ML approaches can be applied to wider domains, which is relevant if the expected location of TC changes with global warming (see our response to Reviewer #1).

We have modified the text of the revised manuscript to make this clearer. The physical algorithms used to find cyclone tracks are based on thresholds calculated using reanalysis data (Bourdin *et al.* 2022). These algorithms are currently applied to simulation data, without changing the value of their thresholds. Of course, this practice is debated in the community and some argue that they should be adapted. It is clear that reanalysis data and climate model simulations represent cyclones in different ways because of model imperfections. We plan to compare deep learning models and physical algorithms that are trained and/or tuned on reanalysis data and applied on climate simulations. In particular we want to know if a deep learning model is able to generalize better than physical algorithms, when using simulation data. However, we felt it was important to study the effect of the learning dataset as a preliminary step towards answering this question. In addition to the changes previously presented, we have added these lines in the Introduction:

"It is common practice that data from climate simulations are first produced and stored, and then analyzed. However the climate modelling community is also moving in the direction of "on-the-fly" (also called in situ) data analysis in order to reduce the volume of data to be stored and the environmental impacts of such storage. This paradigm change implies the development of more efficient analysis methods. Both physical algorithms (i.e., trackers) and DL models are legitimate approaches to study TCs but it is useful to understand if one generalizes better than the other. However, it is important to understand that both approaches do not necessarily achieve the same thing. Indeed, trackers search for the trajectory of a cyclone by detecting its different positions in time, whereas the DL models listed in Table 1, derived from computer vision, detect cyclones on an image frozen in time. Other DL approaches are available for object tracking."

- (L41-54) Several ML studies has been cited in this paragraph. However, the current manuscript does not clearly present the weakness (or strength) of these previous studies over this study. Since this study is extending the ML approach side of TC detection algorithms, the previous studies (in terms of their algorithms and performance) need to be introduced with more details such as which aspect of these studies are limitations and how this study improves on those points.

This point was raised by Reviewer #1 as well. We have summarized in a Table the neural network architectures that have been used to perform TC classification, detection and segmentation over the last 8 years.

Since Liu *et al.* (2016), whose deep learning model only classifies pre-existent images, various subsequent studies have focused on improving the detection of all cyclones at once present in unidimensional or multidimensional meteorological images (e.g., Ebert-Uphoff and Hilburn, 2020) and climate model data (e.g., Matsuoka et al., 2018). The detection can be either "coarse" by drawing rectangular envelopes around the cyclones ((the studies are flagged as detection in the Purpose column of the Table), or "precise" by drawing the contours of the cyclones including their internal structure (studies flagged as segmentation). The main idea of these more recent studies is to apply new deep learning model architectures coming from computer vision research (e.g., U-Net, DeepLabv3, YOLOv3, Single Shot Detector, etc.) to the analysis of meteorological features such as cyclones.

This has given us the opportunity to clarify our motivations in the Introduction section of the revised manuscript. To our knowledge, no study has investigated the impact of interpolation on the performance of a CNN or compared the performance of a CNN trained on different image datasets with a statistically more robust method.

We have chosen a cyclone image classification approach mainly to avoid the difficult problem of data labeling required for cyclone detection or segmentation and the overfitting of these methods when the dataset is of modest size.

It is also interesting to note that we provide a new hint on the superior information richness of ERA5 over MERRA-2 for cyclone detection, which was strongly suggested by the study of Malakar *et al.* (2020) who conclude that ERA5 provides the best representation of the TC structure in terms of intensity, out of five reanalysis including MERRA-2.

We have added the following lines in the Section 1:

"There is also a wealth of studies on the detection of TCs in satellite imagery, reanalysis and climate model outputs based on machine learning (ML) approaches. Table 1 summarizes notable studies published in the last eight years that implement neural architectures based on convolution layers. It is not surprising that this approach was favored because TCs have very distinct features which make them relatively easy to detect with convolutional neural networks. Since Liu et al. (2016), whose deep learning (DL) model only classifies pre-existent images, various subsequent studies have focused on improving the detection of all cyclones at once present in unidimensional or multidimensional meteorological images (e.g., Ebert-Uphoff and Hilburn, 2020) and climate model data (e.g., Matsuoka et al., 2018). This latter work focuses on the detection of cyclones using a CNN image classifier which operates on a sliding window of output from the Nonhydrostatic Icosahedral Atmospheric Model (NICAM) and studies the system performance in terms of detectability. The detection can be either "coarse" by drawing rectangular envelopes around the cyclones (the studies are flagged as detection in the Purpose column of the Table) or "precise" by drawing the contours of the cyclones including their internal structure (studies flagged as segmentation). The main idea of these more recent studies is to apply new DL model architectures coming from computer vision research (e.g., U-Net, DeepLabv3, YOLOv3, Single Shot Detector, etc.) to the analysis of meteorological features such as cyclones."

| Authors | Year | Purpose | Dataset | Variables | NN Architecture | Performance at best |
|---------|------|---------|---------|-----------|-----------------|---------------------|
| Liu *et al.* | 2016 | Cyclone image classification | ERA-interim (rea) CAM5.1 (mod) NCEP-NCAR (rea) 20th Century Reanalysis | Pressure sea level, wind vectors at 10 meters and at 850 hPa, temperature at 200 and 500 hPa, total column water vapour | ad hoc CNN | Accuracy: 99 % |
| Racah *et al.* | 2016 | Cyclone segmentation | CAM5 (mod: 25 km) | The 16 channels of CAM5 | ad hoc autoencoder | mAP@IoU=0.1: 52.92 % |
| Hong *et al.* | 2017 | Eye detection | COMS-1 (sat) | 4 IR channels | GoogLeNet | RMSE: 0.02 |
| Matsuoka *et al.* | 2018 | Detection of cyclone by sliding window and cyclone image classifier | NICAM (mod: 14 km) + NSW6 + HadISST | Outgoing longwave radiation | ad hoc CNN | Probability of de tection: 79.9–89.1 % false alarm ratio: 32.8–53.4 % |
| Kumler-Bonfanti *et al.* | 2020 | Cyclone segmentation | GFS (mod) | Total precipitable water | U-Net | Accuracy: 0.991 Dice coeff: 0.763 Tversky coeff: 0.75 |
| Shakya *et al.* | 2020 | Cyclone image classification | KALPANA-I (sat) MOSDAC (sat) | IR, Vis | ad hoc CNN | Accuracy: 97 % |
| Shakya *et al.* | 2020 | Cyclone detection and path prediction | KALPANA-I (sat) MOSDAC (sat) | IR, Vis | RetinaNet and polynomial regression | RMSE: 5-15.55 % |
| Prabhat *et al.* | 2020 | Cyclone segmentation | CAM5.1 (mod: 25 km) | Integrated vapor transport, integrated water vapor, vorticity, | DeepLabv3+ | IoU: 0.2441 |

| | | | | wind vectors at 10 meters and at 850 hPa, sea level pressure | | |
|---|---|---|---|---|---|---|
| Pang *et al.* | 2021 | Cyclone detection | Satellite images from NII | Vis | DCGAN and YOLOv3 | Accuracy: 97.78 % mAP@IoU=0.5: 81.39 % |
| Shi *et al.* | 2022 | Extratropical cyclone detection | ERA5 (rea) | Top net thermal radiation, mean sea level pressure, vorticity | Single Shot Detector | mAP@IoU=0.5: 79.34-86.64 % |

Table. Summary of previous studies aiming at the detection or the segmentation of TC in satellite or model data. coeff: coefficient; IoU: intersection over union; IR: infrared; mAP: mean average precision; mod: model; rea: reanalysis; RMSE: root mean square error; sat: satellite; vis: visible.

Additional reference:

Malakar, P., Kesarkar, A. P., Bhate, J. N., Singh, V., & Deshamukhya, A. (2020). Comparison of reanalysis data sets to comprehend the evolution of tropical cyclones over North Indian Ocean. *Earth and Space Science*, 7, e2019EA000978. doi: 10.1029/2019EA000978

- L49: "lack sufficient details": elaborate "details"

As far as we can see, the published studies do not especially describe in detail the data engineering involved for training and evaluating models in deep learning applied to climate sciences. Moreover, not all publications describe the evaluation methods of the metrics used to measure the models. Thus, we have shed light on the preparation of the image sets, described the background image generation algorithm which seeks to avoid at all costs any correlation with the cyclone images and background images already generated, while seeking to maximize the classification decision boundaries. We also described precisely our iterative cross validation method that allowed us to evaluate the uncertainty on the values of the model performance metrics during our numerical experiments. Finally, we would like to caution against the commonly used accuracy metric, which has a bias introduced by its class threshold.

We have modified the manuscript to discuss this:

"While such techniques are now mainstream, they are not always well documented and their description may lack sufficient details which are often key in ML, like the data engineering involved in the preparation of the training dataset, the hyperparameters of the CNN and the evaluation methods of the metrics used to measure the performance of the models."

- L75-76: Explain why tropical depressions are not included as TCs. Is the performance of the current CNN algorithm sensitive to the inclusion of tropical depressions as TCs?

We chose to exclude tropical depressions (corresponding to the HURDAT2 status "TD") as well as the other weather events of lower intensity, because we wanted to focus on intense cyclonic events (> 34 knots) which are responsible for the largest impacts when they reach land. Our final objective is to detect cyclone in climate simulations.

This said we believe that the CNN would be sensitive to the characteristics of tropical depressions. As mentioned in the section "potential future work", a ternary classification experiment can be envisaged by adding the tropical depression class in the training dataset.

We have modified the manuscript to explain why we choose to exclude TDs in Section 2.1 on TC dataset:

"We chose to exclude tropical depressions (corresponding to the HURDAT2 status TD) as well as the other weather events of lower intensity, because we wanted to focus on intense cyclonic events (> 34 knots) which are responsible for the largest impacts when they reach land."

We also discuss a multiclass classification (including extra tropical cyclones) in the Discussion section:

"We have chosen a binary approach for the classification (i.e., TC or background), but it is quite possible to design a classifier predicting the range of HURDAT2 status of the images. Such a classifier would use nine neurons with the soft max activation function as the last layer of the CNN. However, training it would probably face an acute problem of image set imbalance. Indeed, four classes out of nine have a number of occurrences smaller than 400 in HURDAT2 (see Fig. A1). To improve the situation, it would be possible to merge some classes (e.g., WV with DB and SD with SS) in order to mitigate the problem."

**Section 2**

- Section 2.2: The difference and similarity between ERA5 and MERRA2 need to be presented in more detail. The atmospheric models and assimilation schemes are surely different, but the observation dataset they use might have good overlaps. How different these two datasets are is important to interpret the results of cross examination. That is, if they are only slightly different, the robust results from cross examination could be rather trivial.

The Reviewer is right that we expect two high-quality reanalyses to be "relatively similar" because they ingest similar observational data. At least the representation of TCs should be more similar in two different reanalyses than in two different climate models. This is probably why we do see robust results from our cross training methods. However there are also differences as we show in the manuscript. The literature also provides some elements for a more thorough discussion.

The representation of TC by ERA5 and MERRA-2 was studied at least by two studies: Malakar *et al.* (2020) and Hodges *et al.* (2016).

The study of Malakar *et al.* (2020) analyzes the error in the location of the center, maximum winds and minimum pressure at sea level in six meteorological reanalyses including ERA5 and MERRA-2 for the evolution of 28 TCs occurring between 2006 and 2015 over the North Indian Ocean, with respect to the observations of the Indian Meteorological Department (IMD). The authors of this study show, among other things, that the ERA5 dataset captures the evolution of these TCs in a more realistic way than MERRA-2 (i.e. smaller errors in the previous variables). They also show that ERA5 and MERRA-2 can capture the intensity of the TCs from the depression stage to the very severe cyclonic storm stage but not the last stage which is the extremely severe cyclonic storm stage: the intensity of the TCs is underestimated. However, they conclude that of the six datasets, ERA5 provides the best representation of the TC structure in terms of intensity.

The study published by Hodges *et al.* (2016) shows that 95 % of the Northern Hemisphere TC tracks, from the IBTrACS database that includes HURDAT2, are present in MERRA-2. Unfortunately, this study does not include ERA5. It also confirms the underestimation of cyclone intensity in MERRA-2 compared to observations.

The conclusion has been modified as in the response to Reviewer #1's comments:

"Applying an ERA5-trained CNN on MERRA-2 images works better than applying a MERRA-2 trained CNN on ERA5 images, which suggests that ERA5 has a larger information content in the framework of our CNN.

This is also consistent with the findings of Malakar et al. (2020) who analyzed the error in the location of the center, maximum winds and minimum pressure at sea level in six meteorological reanalyses including ERA5 and MERRA-2 for the evolution of 28 TCs occurring between 2006 and 2015 over the North Indian Ocean, with respect to the observations of the Indian Meteorological Department (IMD). The authors of this study show, among other things, that the ERA5 dataset captures the evolution of these TCs in a more realistic way than MERRA-2 (i.e., smaller errors in the previous variables). They also show that ERA5 and MERRA-2 can capture the intensity of the TCs from the depression stage to the very severe cyclonic storm stage but not from the extremely severe cyclonic storm stage for which the intensity of the TCs is underestimated. However, they conclude that of the six datasets, ERA5 provides the best representation of the TC structure in terms of intensity. Finally, the study published by Hodges et al. (2017) shows that 95 % of the Northern Hemisphere TC tracks, from the IBTrACS database that includes HURDAT2, are present in MERRA-2. Unfortunately, this study does not include ERA5. It also confirms the underestimation of cyclone intensity in MERRA-2 compared to observations."

Additional reference:

Hodges, K., Cobb, A., & Vidale, P. L. (2017). How well are tropical cyclones represented in reanalysis datasets?, *Journal of Climate*, *30*(14), 5243-5264.

**Section 3**

- Section 3.2.6: I assume we are using bi-linear interpolation here?

Indeed, we applied a bi-linear interpolation. This information will be added in the revised manuscript.

- L226-240: Rewrite. The authors' choice (iterative cross-validation) was introduced in a confusing way (after hold-out and k-fold). Consider starting the paragraph with mentioning iterative cross-validation and then explaining its benefit over k-fold.

We have rewritten the description of the evaluation of the metrics as following:

"For the evaluation and comparison of the metrics (developed in Section 4.2), we wanted to be able to calculate the expected value and the uncertainty of the metrics, without bias. To that end, we applied an iterative cross validation method which consists in repeating 20 times a cross validation method. We chose the k-fold method (Bishop, 2006), with k equal to ten, as the cross validation method. We obtained a mean of the metrics for each k-fold

iteration. By applying the central limit theorem on this set of metric means, we could compute the expected value and the uncertainty of the metrics.

In order to avoid any bias, we took care to check if the central limit theorem can be applied, by testing the normality of the distribution of the metric means using the Shapiro-Wilk statistical test (brief non-mathematical presentation given in Appendix B1). Moreover, images coming from a time series of tracks from the same cyclone may be found in both the training and test datasets, which would induce some dependance between the training and test datasets due to the autocorrelation within individual cyclone tracks. In order to avoid such a bias, the k-fold split is based on sampling the years randomly and balancing the folds as much as possible. The partitioning combinations are calculated in advance in order to guarantee the uniqueness of their composition. Scale bias is also avoided by standardizing the channels of the images online, just before training the CNN.

Finally, for the comparison of the metric means, we chose to apply the Kruskal-Wallis statistical test (brief non-mathematical presentation given in Appendix B2) for an alpha level of 1 %, because the Shapiro-Wilk test was negative for most distributions of metric values of our experiments, invalidating the use of the Student's t-test.

For the experiment of highlighting the problem with the accuracy (point developed in Section 4.1), we applied the classical hold-out method, avoiding the autocorrelation between images belonging to a same cyclone track, with the following partitioning: 70 % of the data for the training dataset and 30 % of the data for the test dataset."

New reference added:

Bishop, C. M., Pattern Recognition and Machine Learning, 1, *Springer International Publishing*, 2006, 32-33

- [3] Lack of details on the methodology of statistical tests and metrics: I do appreciate the authors' efforts to use more robust statistical metrics and tests. I believe the following metrics and tests are less common (unlike 'accuracy'): AUC, AUPRC, Shapiro-Wilks test, and Kruskal-Wallis test. Including brief mathematical descriptions for those metrics and tests—maybe in the appendix—would be a great service to readers (including myself) who are not familiar with them.

We have briefly described the usefulness of the AUC and AUPRC metrics in Section "3.4 Evaluation of metrics". As the Reviewer points out, it is necessary to add some explanations of these notions with some Figures. Here is what we have added to the revised manuscript:

"The AUC measures the power of a model to discriminate between the two classes for a variety of decision threshold values. The AUC is, as the acronym indicates, the area under the ROC curve which is plotted by the values that recall (the ability of a model to identify all occurrences of a class) and false positive rate of a model take (definitions are given in Appendix B1). The recall and false positive rate values are calculated according to the ground truth and the classifier responses for a given test dataset and for all possible decision thresholds (or a set of discrete values). A perfect classifier has an AUC equal to 1, recall all the images of cyclone with a null false positive ratio.

The AUPRC measures the recall of a model while minimizing the precision (prediction errors). The AUPRC follows a similar approach to the AUC: it is the area under the curve which is plotted by the values that the precision and recall of a model take (for all possible decision thresholds, etc.). A perfect classifier has an AUPRC equal to 1, recall all the images of cyclone without wrongly classifying any background image as a cyclone image."

We also have explained in Appendix the usefulness of the Shapiro-Wilks and Kruskal-Wallis statistical tests, as well as a simple explanation of how they work, without going into mathematical details that we find a bit irrelevant:

"Let us consider a group, a set of values of a random variable that we observe, for example during an experiment, and its population, the set of all the values that the variable can take for a particular experimental context. The group is a subset of the population.

The Shapiro-Wilks test poses the so-called null hypothesis (H0) that the group of values that a given quantitative variable takes, comes from a normally distributed population. For a non-technical explanation of the test, we can say that the Shapiro-Wilks test quantifies in a single metric, the p-value, the dissimilarities between the distribution of the values of the group and the distribution of the population of the variable if it was normal.

For a risk of error called alpha level, commonly fixed at 1 % or 5 %, H0 null is rejected if the p value is lower than the alpha level and is accepted if the p value is higher than the alpha level. The latter represents the risk of accepting H0 when it is not true: a false positive. If H0 is accepted, random sampling of the group can explain the dissimilarities between the distribution of the values of the group with a normal population distribution. If H0 is rejected, it can be stated that the population of the variable is not normally distributed.

The comparison of the mean of the values of a variable from two different groups is usually done by the Student's t-test (or two-sample ANOVA) or the Kruskal-Wallis test. The Student's t-test is a parametric statistical test, in this case it requires that the distribution of the given variable is normal. The Kruskal-Wallis test is non-parametric and does not require the assumption of the population distribution of the variable. Indeed, it is not based on the value that the variable takes, but on its rank in the classification of the observed values of the variable. As for the Shapiro-Wilks test, we propose a non-technical explanation of the Kruskal-Wallis test: this test quantifies in a single metric, the p-value, the dissimilarities between the mean ranks computed for two or more groups of values, with, as H0, random sampling of the group being able to explain the differences between the medians of the two groups, because they may come from the same population."

- Insufficient details of the authors' ML methodology: I suggest adding more details about how the authors implemented in the *main* text.
  - Move Tables A2 and A3 to the main text. These are key information for ML implementations. Since this paper evolves around ML, these bits of information should be included in the main body of the manuscript.

We agree with the Reviewer and have moved Tables A3 and A2 to Section 3.3.

- Table 1: Add information about stride and padding for convolution and pooling layers. Readers can infer them from the values in the table, but it is better explicitly shown. Also, specify pooling was done using either 'max' or 'averaging'.

We agree with the Reviewer: we will add in the caption of the Table that there was no padding added and that the stride was (1, 1). We will also specify that pooling is max pooling.

- Table A3: Add a column that shows the search space for hyperparameter search.

We agree with the Reviewer: we have added the following information as a new column in Table A3:

| Hyperparameter | Value | Search space |
| --- | --- | --- |
| Loss function | Binary cross-entropy | - |
| Training metric | Loss computed on test set | - |
| Maximum number of epoch | 100 | - |
| Early stopping number of epoch | 10 | - |
| Batch size | 256 | from 32 to 256, step of 32 |
| Optimizer | Adam | Adam; SGD |
| Learning rate | 0.0001 | 0.0001; 0.001; 0.01 |

- The hyperparameter optimization done in this study does not include some of the important hyperparameters, $g$., the number of layers, the number of neurons per layer, and the number of filters (convolution). It is likely that these hyperparameters would have impact on the performance of the CNN model more strongly than those listed in table A3. Provide justification on why these hyperparameters were omitted during the hyperparameter search process.

The authors agree with the Reviewer: our hyperparameter optimization does not include the architecture of the CNN. First of all, we would like to clarify our intention regarding the optimization of the hyperparameters: our manuscript is based on the paper written by Liu *et al.*, but these authors do not mention the values of their training hyperparameters, i.e. batch size, optimizer and learning rate. Instead of fixing these values in an arbitrary way, we

search for local optimal values of these hyperparameters to maximize the performance of the CNN.

The following lines have been added to Section 3.1 Classification model:

"We did not try to optimize the architecture of the CNN proposed by Liu *et al.* because we believe they have already optimized it and the modifications we have made do not require any further optimization, since the performance of our CNN is very close to that of Liu *et al.* Finally, as explained above, we prefer to focus on the performance and the sensitivity of the CNN to the learning dataset instead of obtaining better performances."

- In Section 3.1 (or somewhere in Section 3), explicitly write how the input vectors were normalized and/or transformed. This information is presented vaguely later in the manuscript (L241-243), but it should be written more clearly, considering input vector normalization is one of the critical factors that determines the performance of ML algorithms.

We agree to mention in Section 3 on model training how the data of each channel taken separately were standardized, as it is a necessity for the training of the CNN and we recall it at line 241. We have added the following section:

"3.2.7 Data standardization

Neural network models learn a mapping from input variables to an output variable. The input variables have nearly always different scales and large scale differences are detrimental to the learning process of neural networks. In order to ensure that each variable is equally important, whatever its range of values, input variables are rescaled to the same scale. There are several methods such as standardization (or Z-score normalization) which consists in recalculating the values of the variables so that their mean and standard deviation equal to zero and one, respectively. In our study, we have systematically standardized each channel of the images, by calculating the means and standard deviations of the channels on all the images of the training set. The validation and test image datasets are excluded from the calculation of the mean and standard deviation, to avoid that information about the validation and test datasets leak into the training phase. However the validation and test datasets are also scaled using the mean and standard deviation of the training dataset."

- L176: Specify which task was for the 135-min CPU time.

It was for the extraction of all the channels of the images. We have added this information: "The CPU time was 135 minutes for the extraction of all the channels of the images."

**Section 4**

- Insufficient comparison of the CNN model performance: Section 4 presents the performance of the author's ML detection algorithm. However, it is hard to tell if the authors' algorithm is good or not because the benchmark comparison with other

state-of-art algorithms is not provided. Only benchmark comparison provided is "simple metrics" in Figure 7; even then, the details about "simple metrics" are not included in the manuscript. Regardless, *simple* metrics are not a fair benchmark for the current study. A better (and more useful) benchmark should be state-of-art TC detection algorithms—including both physical and ML algorithms—that has been presented in recent years (*e.g.*, some of previous studies listed in L31 and L42). In particular, Liu *et al.* (2016) which this study is based on, would provide an objective baseline. The suggested comparison will further enhance the merit of the authors' work.

We agree with the Reviewer and have added the previous Table that sums up the experiences of detecting cyclones in meteorological images. It can be noted that the accuracy of our CNN is in the same range as Liu *et al.* (2016) with 99 % and Shakya *et al.* (2020) with 97 %.

We also have added these explanations about the simples metrics:

"Indeed, the usefulness of a model is measured by the difference between its performance and that of models based on simple rules or a domain specific baseline. For instance, we implement the following simple models (from the software library scikit-learn): "most frequent" which always predict the most frequent class observed in the training dataset (i.e., background), "stratified" which generates randomly predictions at probabilities that respect the class distribution of the training dataset (i.e., 1/3 cyclone, 2/3 background), "uniform" which generates predictions uniformly at random background or cyclone with equal probability."

We removed the "constant" and "prior" models that behave identically to "most frequent" in our experiment.

About the performance of algorithms based on physical equations, we can cite the prepublication by Bourdin *et al.* (2022) who performed an intercomparison of four cyclone trajectory detectors - called trackers - on the same ERA5 data. The authors report that the algorithms have a recall that ranges between 75 and 85 % and a false discovery rate between 9 and 36 % after post processing. However, it is important to understand that trackers do not have the same objectives as the DL models of classification, detection and segmentation listed in the previous Table. Indeed, trackers search for the trajectory of a cyclone by detecting its different positions in time, whereas DL models, derived from computer vision, detect cyclones on an image at a given point in time.

The following lines have been added in the Introduction:

"Bourdin et al. (2022) performed an intercomparison of four cyclone trajectory detectors –called trackers– on ERA5 reanalysis. [...] Both physical algorithms (i.e., trackers) and DL models are legitimate approaches to study TCs but it is useful to understand if one generalizes better than the other. However, it is important to understand that both approaches do not necessarily achieve the same thing. Indeed, trackers search for the trajectory of a cyclone by detecting its different positions in time, whereas the DL models

listed in Table 1, derived from computer vision, detect cyclones on an image frozen in time. Other DL approaches are available for object tracking."

- L216: What was the main criticisms, offered by Provost *et al.* (1997) and Ling *et al.* (2003), to the accuracy metric?

The articles by Ling *et al.* (2003) and Provost *et al.* (1997) mainly concern a comparison between AUC and accuracy and suggest favoring AUC over accuracy.

The main criticism of Ling *et al.* (2003) towards the accuracy is about the loss of information it induces during the transformation of the probability, returned by the classifier, into a class identifier: as soon as this probability exceeds the class decision threshold, the response of the classifier takes the class identifier while the information of the difference between the value of the probability and the threshold is lost. The authors also report a loss of discrimination power on an unbalanced dataset. Note that while AUC and AUPRC also transform probabilities into class identifiers, these metrics retain the probability information, as they are computed for all threshold values.

Provost *et al.* (1997) propose two fundamental justifications to explain the use of accuracy as a machine learning model metric and show that both justifications are questionable at best. The first justification states that "The classifier with the highest accuracy may well be the classifier that minimizes cost, especially when the classifier tradeoff between true positive and false positive predictions can be resolved." By cost, the authors mean the cost of missed classification, i.e., producing false positives and false negatives. As a reminder, depending on the context of the classification, a false negative can have very strong consequences (e.g. a medical test). The authors recall that accuracy assumes equal costs of misclassification. The second justification states that "The induction algorithm that produces the most accurate classifiers can also produce minimum cost classifiers by training it differently."

For the first justification, the authors argue that in real world cases, the distribution of classes is generally not known. On the other hand, the authors also point out that it has been proven that it is virtually impossible to assign the misclassified costs precisely. Thus, the authors show that justification 1 is generally not verified, because it is impossible to optimize the misclassified rate.

For the second justification, the authors use the properties of ROC curves whose area is the value of the AUC. ROC curves allow to visually show that a classifier "dominates" another classifier with its ROC curve strictly above the other classifier's. The authors state that "a dominating model is at least as good as all other models for all possible cost and class distributions".The authors demonstrate on some examples that ROC curves, produced by the averaging 10-fold cross-validation method, are statistically intertwined without being able to declare one classifier dominating the others, while their accuracy is not significantly different.

We have summarized this explanation into this following sentence added in Section 4.1 on "Accuracy and its threshold":

"Accuracy is a convenient measure, but according to Provost et al. (1997) and Ling et al. (2003), the class threshold makes it non objective. Provost et al. (1997) argue that in real world cases, the use of accuracy as an ML model metric is questionable at best, because the distribution of classes is generally not known so it is impossible to optimize the misclassified rate. Furthermore, the authors demonstrate in some examples that classifiers, having their accuracy not significantly different, are not equivalent concerning the AUC. Ling et al. (2003) argue that the accuracy losses information during the transformation of the probability, returned by the classifier, into a class identifier: as soon as this probability exceeds the class decision threshold, the response of the classifier takes the class identifier while the information of the difference between the value of the probability and the threshold is lost."

- L256: Provide the usefulness of Youden's index, as done for AUC and AUPRC in L217-219

We have added some hints about the usefulness of Youden's index:

"Then we calculated the decision threshold for which the number of misclassified predictions is minimal. For this purpose, we use the Youden's index which is a measure of the tradeoff between sensitivity and specificity. Maximizing the index means minimizing the false positives and false negatives according to its equation described in Appendix C6, knowing that Youden's index varies according to the class decision threshold."

- Figure 8: The convention of box plots is standard, but I suggest including a brief explanation about what whiskers, boxes, and diamond markers stand for in the figure legend.

We have included a short explanation of the whisker plots in the caption:

"Box plots are a synthetic representation of a data distribution. They are composed of a box and two whiskers. The bottom of the box corresponds to the first quartile ($Q_1$) of the studied dataset, below which 25 % of the data are located. The middle line of the box is the median of the dataset, and the top line of the box corresponds to the third quartile ($Q_3$), below which 75 % of the data are located. The interquartile range is represented by the extent of the box: $IRQ = Q_3 - Q_1$. The bottom whisker is calculated according to the IRQ: $Q_1 - 1.5 \times IRQ$ and the top whisker: $Q_3 + 1.5 \times IRQ$. The data outside the whiskers are considered as outliers, and represented as diamond markers."

**Section 5**

- L288-290: The authors' conclusion seems premature. There are other possibilities, for example, the current CNN architecture might be somehow more suitable for ERA5 dataset.

This point was also made by Reviewer #1 and we agree that our conclusion should be slightly qualified as it may not generalize to other CNN architectures. We would like to clarify

our conclusion about MERRA-2 having less information content for TC image classification than ERA5:

"Applying an ERA5-trained CNN on MERRA-2 images works better than applying a MERRA-2 trained CNN on ERA5 images, which suggests that ERA5 has a larger information content in the framework of our CNN.

This is also consistent with the findings of Malakar *et al.* (2020) who analyzed the error in the location of the center, maximum winds and minimum pressure at sea level in six meteorological reanalyses including ERA5 and MERRA-2 for the evolution of 28 TCs occurring between 2006 and 2015 over the North Indian Ocean, with respect to the observations of the Indian Meteorological Department (IMD). The authors of this study show, among other things, that the ERA5 dataset captures the evolution of these TCs in a more realistic way than MERRA-2 (i.e., smaller errors in the previous variables). They also show that ERA5 and MERRA-2 can capture the intensity of the TCs from the depression stage to the very severe cyclonic storm stage but not from the extremely severe cyclonic storm stage for which the intensity of the TCs is underestimated. However, they conclude that of the six datasets, ERA5 provides the best representation of the TC structure in terms of intensity. Finally, the study published by Hodges *et al.* (2017) shows that 95 % of the Northern Hemisphere TC tracks, from the IBTrACS database that includes HURDAT2, are present in MERRA-2. Unfortunately, this study does not include ERA5. It also confirms the underestimation of cyclone intensity in MERRA-2 compared to observations."

**Technical comments**

**1:**

- Line 10: specify spatial interpolation
- Figures 4, 5, 10, and 11 are missing units. If they are dimensional you could add colorbars, if they are all standardized you could mention it in the caption.
- Line 7: "similar locations and times to the TC containing images"
- Line 8: "accuracy, but"
- Line 9: "activity, but"
- Line 19: "causes"
- Line 24: "Indeed,"
- Line 46: "output from the Nonhydrostatic"
- Line 93: why "usually"?
- Line 94: "relevant" is an odd word choice, maybe "appropriate"?
- Line 98: "are able to capture better" --> "better capture"
- Line 101: strike "So"
- Line 107: add a comma after "indeed"
- Line 148: "a third" --> "one third"
- Line 276: Figure number is missing
- Line 287: change "experienced"
- Line 287: strike "well"
- Line 241: "image" --> "images"
* * *
**2:**

- L45: This later work → This latter work
- L83-84: Specify if 0.25x0.25 and 0.5x0.6 is either native or interpolated resolutions. If it is the latter, specify which interpolation scheme was used to regrid.
- L130-131: Rewrite the sentence: "thought", "those", and "they" are unclear.
- L134: Provide a proper citation for UML2.
- L203: Specify N. Is it 10 (Table A3)?
- Figures 4, 5, and 10: include colorbars.
- Figure 6: Specify the y-axis uses a logarithmic scale in the figure legend.
- Figure7: Define the x-axis tick labels in the figure legend, as done in Figure 9
- Figure 9: 'purple' looks more similar to red.
- L324: Figure number is missing.

We have implemented these technical corrections.

---

## Author Response (AR2)

Dear Reviewers,

We would like to thank you again for your contribution to the improvement of our article. We make below a point-by-point answer to the Reviewer's last remarks.

We wish you a good receipt of our modified manuscript.

Regards,

The authors

**Reviewer 2**

- I believe the following reorganization would increase the readability of this manuscript. First, I suggest moving L470-480 to Section 5 Discussion since it is proper material for discussion. Besides, having such material in a conclusion section distracts readers and dilutes the final points of the manuscript.

We agree with the Reviewer. The paragraph has been moved to Section 5.

- Second, this is optional, but I suggest the authors consider moving most (of all) of appendix tables to a supporting material. These are important bits of information, but not critical for understanding the main texts. Note that this manuscript already has six tables.

We could agree with the Reviewer, but we would like to seek the Editor's advice before moving the tables.

- There are numerous typos and unclear expressions in the newly added text. These are not comprehensive; I urge the authors to read through again to polish unrefined bits.

- L25: "future expected changes" to "future changes"
- L30: Add commas before "but" and after "Furthermore"
- L32: Add commas before and after "thus"
- L53: "pre-existent images" is vague; elaborate.
- L59: "the Table" to "Table 1"
- L64: "like" to "e.g.,"
- L78-79: Give examples of "Other DL" with proper citations, or remove the sentence.
- L153: "homogeneous" is vague; elaborate.
- L175: "OMG" to "Object Management Group"
- L238: "an output variables" to "output variables"
- L240: "whatever" to "regardless of"

- L245: Add a comma after "However"
- L253: "lost function" to "loss function"
- L265: "for" to "with"
- L289: Add "that is," before "recall"
- L334-335: The sentence starting with "Furthermore" is unclear; rewrite.
- L335: "losses" to "loses"
- L450: Add a comma after "However"
- L495: Remove the comma after "takes"
- L497 and L509: "in a single metric, the p-value" to ", using a single metric (the p-value)"

We have implemented these technical corrections.

- L360, L426, and L441: "not an issue" and "very high" are vague. Consider rephrasing to "competent to other existing methods"

In line 360, we have added a comparison of metric values with the two other studies about image classification that we have listed in Table 1. For lines 426 and 441, we just recalled that the value of AUC and AUPRC are over 0.99.

The following lines have been added:

"The values of the metrics are very high, over 0.98. They are close to those reported in the studies about image classification that we have listed in Table 1. As a comparison, the accuracy value of our CNN is between those of Shakya et al. (2020) and Liu et al. (2016): 0.97 < 0.98 < 0.99. However, this comparison should be put into perspective: our method of calculating the accuracy is more robust against uncertainty. Additionally, the model of Shakya et al. (2020) is trained and tested on observational data that are quite different from multidimensional meteorological reanalysis data."

- L328: Is a 70:30 split correct? In L264, the split used for hyperparameter tuning is 85:15 since the validation dataset should be considered as a part of training dataset. Just to double check.

The split is correct because we consider that the validation dataset is part of the test dataset as we wanted to detect any overfitting situations.

- L445-447: Needs citations of studies that have shown the more frequent "situations."

We were referring to the studies presented in the introduction. We now mention it in parenthesis.